# Integrative Taxonomy and Molecular Phylogeny of the Plant-Parasitic Nematode Genus *Paratylenchus* (Nematoda: Paratylenchinae): Linking Species with Molecular Barcodes

**DOI:** 10.3390/plants10020408

**Published:** 2021-02-22

**Authors:** Phougeishangbam Rolish Singh, Gerrit Karssen, Marjolein Couvreur, Sergei A. Subbotin, Wim Bert

**Affiliations:** 1Nematology Research Unit, Department of Biology, Ghent University, K.L. Ledeganckstraat 35, 9000 Ghent, Belgium; g.karssen@nvwa.nl (G.K.); marjolein.couvreur@ugent.be (M.C.); wim.bert@ugent.be (W.B.); 2National Plant Protection Organization, Wageningen Nematode Collection, P.O. Box 9102, 6700 HC Wageningen, The Netherlands; 3Plant Pest Diagnostic Center, California Department of Food and Agriculture, 3294 Meadowview Road, Sacramento, CA 95832, USA; sergei.a.subbotin@gmail.com; 4Center of Parasitology of A.N. Severtsov Institute of Ecology and Evolution of the Russian, Academy of Sciences, Leninskii Prospect 33, 117071 Moscow, Russia

**Keywords:** D2-D3 of 28S, ITS, 18S, *COI*, morphology, morphometrics, *Paratylenchus*, plant-parasitic nematodes, phylogeny, taxonomy

## Abstract

Pin nematodes of the genus *Paratylenchus* are obligate ectoparasites of a wide variety of plants that are distributed worldwide. In this study, individual morphologically vouchered nematode specimens of fourteen *Paratylenchus* species, including *P. aculentus, P. elachistus, P. goodeyi, P. holdemani, P. idalimus, P. microdorus, P. nanus, P. neoamblycephalus, P. straeleni* and *P. veruculatus*, are unequivocally linked to the D2-D3 of 28S, ITS, 18S rRNA and *COI* gene sequences. Combined with scanning electron microscopy and a molecular analysis of an additional nine known and thirteen unknown species originating from diverse geographic regions, a total of 92 D2-D3 of 28S, 41 ITS, 57 18S rRNA and 111 *COI* new gene sequences are presented. *Paratylenchus elachistus*, *P. holdemani* and *P. neoamblycephalus* are recorded for the first time in Belgium and *P. idalimus* for the first time in Europe. *Paratylenchus* is an excellent example of an incredibly diverse yet morphologically minimalistic plant-parasitic genus, and this study provides an integrated analysis of all available data, including coalescence-based molecular species delimitation, resulting in an updated *Paratylenchus* phylogeny and the corrective reassignment of 18 D2-D3 of 28S, 3 ITS, 3 18S rRNA and 25 *COI* gene sequences that were previously unidentified or incorrectly classified.

## 1. Introduction

The plant-parasitic nematode (PPN) genus *Paratylenchus* Micoletzky, 1922, commonly known as pin nematodes, are obligate ectoparasites of a wide variety of plants, including herbs, shrubs and trees, that are distributed worldwide and cause various symptoms in their host plants [1,2,3,4,5]. This genus was reviewed by Tarjan [6], who provided the first key to the species. In subsequent years, several attempts were made to split the genus and group its representatives into new genera. The genus *Gracilacus* Raski, 1962, was proposed for members of the *Paratylenchus* species with stylet lengths longer than 48 µm [7]. The validity of *Gracilacus* was supported by Thorne and Malek [8], Raski and Luc [9], Maggenti et al. [10], Raski [11], Esser [12], Andrássy [13] and Yu et al. [14], while Siddiqi [15] treated it as a subgenus of *Paratylenchus*. *Gracilacus* was synonymised with *Paratylenchus* by Brzeski [16], and it was recognized in further works of Siddiqi and Goodey [17], Geraert [18], Brzeski [19], Nguyen et al. [20], Decraemer and Hunt [21], Van den Berg et al. [22], Ghaderi et al. [23], Hesar et al. [24] and Maria et al. [25]. The genus *Paratylenchoides* Raski, 1973 was assigned to *Paratylenchus* species with stronger cephalic sclerotisations, dorso-ventrally narrower heads and small narrow rounded protrusions on the anterior surface of conoid lip region [26]. Siddiqi [15] subsequently lowered *Paratylenchoides* to a sub-generic level for *Paratylenchus*, while Raski and Luc [9] synonymised the two genera owing to the apparent lack of morphological differences between them and Siddiqi [2] accepted this. It was proposed that another genus, *Gracilpaurus* Ganguly and Khan, 1990, included four species displaying long stylets and tubercles on the surface of the cuticle [27]. However, Brzeski [19] did not consider cuticular ornamentation as a generic characteristic, a decision that led to the synonymising of *Gracilpaurus*. The monotypic genus *Cacopaurus* Thorne, 1943 was also proposed and distinguished from *Paratylenchus* by the obese female body, tubercles on annuli of the female cuticle and sessile parasitism [28]. Although Goodey [29] synonymised *Cacopaurus* with *Paratylenchus* due to the lack of consistent differential traits—apart from the female of the former sometimes being sessile and slightly swollen—*Cacopaurus* has been, nevertheless, accepted by Raski [7], Raski and Luc [9], Ebsary [30], Raski [11], Brzeski [19], Siddiqi [2], Andrassy [13] and Ghaderi et al. [23,31]. 

Nematodes of the genus *Paratylenchus* in a broad sense or *sensu lato* are characterised by: small size (<0.7 mm); females being vermiform to obese; C, J or 6 shapes when heat relaxed; two to four lateral lines; cuticle with or without ornamentations; often continuous cephalic regions of rounded to conoid, truncate or trapezoid shapes; protruding or non-protruding submedian lobes; stylet lengths ranging between 10 and 120 µm; well-developed valves of median bulb, slender isthmuses and rounded to pyriform end bulbs in female pharynges; secretory-excretory pores are often at the level between median bulb and end bulb; spermathecae with or without sperm cells; commonly swollen prevulval region; vulvae with or without lateral flaps; presence or absence of a short post-vulval uterine sac; tails ranging from conoid to hemispherical with variable tail termini. The diagnostic traits of juveniles and males are less frequently used for identification, except for looking for the presence of a stylet and looking at the length of the spicules of males. 

Recently, Ghaderi et al. [23] recognized 117 species of *Paratylenchus sensu lato* (excluding *Cacopaurus*), six species of *inquirendae* and four of *nomina nuda*. The nominal species were pragmatically divided into eleven groups based on stylet length, number of lateral lines and absence vs. presence of vulval flaps in females. Since then, seven more species of *Paratylenchus* have been described and linked to DNA sequences [14,25,32,33,34,35,36]. Molecular work on this nematode group is gaining momentum and provides an attractive solution to difficulties encountered in species identification, as well as phylogenetic relationships among species. Subbotin et al. [37], Chen et al. [38,39] and van Megen et al. [40] started to molecularly characterise some *Paratylenchus* spp. using the D2-D3 of 28S rRNA, ITS rRNA and 18S rRNA gene sequences, respectively. Lopez et al. [41] used ITS rRNA gene to examine phylogenetic relationships among four nematode genera; two of the included genera were *Paratylenchus* and *Gracilacus*. Van den Berg et al. [22] conducted the first comprehensive phylogenetic study including several *Paratylenchus* spp. by using 58 28S rRNA and 40 ITS rRNA gene sequences. Several other studies provided additional molecular characterisations, phylogenetic analyses and descriptions of new *Paratylenchus* species [14,25,32,33,34,35,36,42,43,44,45,46,47,48,49,50]. In a study by Hesar et al. [24], 28S rRNA and ITS rRNA gene sequences of several *Paratylenchus* spp. as well as *Cacopaurus pestis* Thorne, 1943, were updated. In addition to providing the first molecular characterisation of *C*. *pestis*, their phylogenetic analyses based on the two partial gene sequences did not support the monophyly of the genera *Cacopaurus*, *Gracilacus* and *Gracilpaurus* that were all found embedded within the clade of *Paratylenchus.*

Despite these recent efforts to integrate and include molecular information in species descriptions and species delineations of *Paratylenchus*, several taxonomic challenges still remain. This is often the case in the field of nematology in general, but the genus *Paratylenchus* is a perfect case in point. Most of the traditionally described species are not yet linked to molecular data, numerous sequences that are currently available are not linked to established species and/or morphological information, sequences are often misplaced and the existence of cryptic species within the genus is common. 

Species boundaries in *Paratylenchus* are sometimes difficult to delimit based solely on morphology because of the limited diagnostic features and morphological plasticity. As of December 2020, only 40 *Paratylenchus sensu lato* species have been linked to molecular data in the GenBank and this database also includes several putative, new, unidentified and incorrectly classified sequences. These misidentified sequences may result in a cascade of erroneous interpretations, including incorrect morphological identification [51] and flawed interpretations of species identity based on relationships in phylogenetic trees. Cryptic species are also likely to represent a component of *Paratylenchus* diversity [22]. It is important to note that correct differentiation of species belonging to agricultural nematode pests from its sibling species has gained importance for a number of reasons, including food security, quarantine regulations and nonchemical pest management strategies [52].

The aims of this study are to: (1) provide and update molecular barcodes of several known and unknown *Paratylenchus* species using four partial sequences—D2-D3 of 28S, ITS and 18S rRNA gene and *COI* gene of mtDNA; (2) link these molecular data to comprehensive morphological information, including light microscopy (LM) and scanning electron microscopy (SEM) images and morphometrics; (3) reconstruct an updated *Paratylenchus* phylogeny; (4) provide a molecular species delimitation for all four markers; (5) reassign unidentified and/or incorrectly classified GenBank sequences to the appropriate species.

## 2. Results

### 2.1. Species Identification, Characterisation and Delimitation

Ten identified and four unidentified *Paratylenchus* species, recovered from soil samples collected in Belgium, were morphologically and molecularly characterised. The identified species were *Paratylenchus aculentus* Brown, 1959, *Paratylenchus elachistus* Steiner, 1949, *Paratylenchus goodeyi* Oostenbrink, 1953, *Paratylenchus holdemani* Raski, 1975, *Paratylenchus idalimus* (Raski, 1962) Siddiqi and Goodey, 1964, *Paratylenchus microdorus* Andrássy, 1959, *Paratylenchus nanus* Cobb, 1923, *Paratylenchus neoamblycephalus* Geraert, 1965, *Paratylenchus straeleni* (De Coninck, 1931) Oostenbrink, 1960 and *Paratylenchus veruculatus* Wu, 1962. The unidentified *Paratylenchus* spp. were *Paratylenchus* sp.2, *Paratylenchus* sp.BE11, *Paratylenchus* sp.D, and *Paratylenchus* sp.F. *Paratylenchus elachistus, P. holdemani, P. idalimus* and *P. neoamblycephalus* were reported for the first time in Belgium and *P. idalimus* was recorded for the first time in Europe. Additional sequences of *Paratylenchus aquaticus* Merny, 1966, *Paratylenchus dianthus* Jenkins and Tylor, 1956, *Paratylenchus hamatus* Thorne and Allen, 1950, *Paratylenchus leptos* Raski 1975, *P. nanus, Paratylenchus projectus* Jenkins 1956, *Paratylenchus shenzhenensis* Wang, Xie, Li, Xu, Yu and Wang, 2013, *P. straeleni* and *Paratylenchus tenuicaudatus* Wu, 1961 and thirteen unidentified *Paratylenchus* species that originated from diverse geographic regions are also provided (Table 1). In total, 68 D2-D3 of 28S, 38 ITS, 57 18S rRNA and 84 *COI* gene sequences were linked to morphological data of the abovementioned ten known and four unknown species collected from Belgium, and 24 D2-D3 of 28S, 3 ITS rRNA and 27 *COI* gene sequences were added to the other nine known and thirteen unidentified species. 

#### 2.1.1. *Paratylenchus aculentus*


*Females* (Sample BE9; Figure 1, Table 2): Heat relaxed specimens open C- to J-shape. Lateral field with three lateral lines. Deirids not observed (not necessarily an indication that they are absent). Cephalic region rounded, low, sometimes appearing slightly truncated, submedian lobes not protruded. Stylet 52–61 µm long, cone 80–91% of stylet length, knobs 2–4 µm across. Pharynx well developed, about one-third of body length. Secretory-excretory pore between median bulb and isthmus level. Spermatheca rounded to slightly oval and filled with sperm cells. Prevulval swelling not prominent. Vulval flaps very small and can be visible under LM. Vulval located at 71–76% of body length from anterior end. Vagina straight to slightly oblique, reaching to almost half of body width. Anus obscure. Tail 18–25 µm long, tapers gradually to a finely or bluntly rounded terminus.

*Molecular characterisation:* Three D2-D3 of 28S, two ITS, two 18S rRNA and four *COI* gene sequences were generated without intraspecific sequence variations. The D2-D3 of 28S and the 18S sequences, respectively, were found to be similar to KP966492 (99% similarity; 4 out of 544 bp difference) and KP966494 (100% similarity; 800 bp) of *P. colinus* from Iran after Hesar et al. [24]. 

*Remarks:* Males were not found. Female morphology and morphometrics matched very well with *P. aculentus*. This species has been reported earlier in Belgium [53]. Although the D2-D3 and 18S sequences pointed towards *P. colinus*, the current population had no cuticular ornamentations present in the anterior part of the body and female bodies were not swollen and submedian lobe protrusions were not seen, which are important characteristics for *P. colinus*. According to Ghaderi et al. [23], *P. aculentus* is part of Group 9 of the *Paratylenchus* species with stylet lengths longer than 40 µm, three lateral lines and absence of vulval flaps. Here, we confirm the presence of small vulval flaps in *P. aculentus*, clearly supported by SEM. This was also an observation originally made by Brzeski [19]. *Paratylenchus aculentus* should, therefore, be placed in Group 8 with *P. colinus* and *P. idalimus*; furthermore, the close affinity of our *P. aculentus* population with *P. colinus* is also molecularly supported by the very conserved 18S rRNA gene fragment.

#### 2.1.2. *Paratylenchus elachistus*


*Females* (Sample BE15; Figure 2, Table 3): Heat relaxed specimens open C- to J-shape. Lateral field with four lateral lines. Deirids visible under SEM. Cephalic region conical-rounded to sometimes truncated. *En face* square-shaped, showing poorly developed submedian lobes, two pronounced lateral ridges and small indistinct dorso-ventral ridges around oral opening, two slit-like amphidial openings laterally. Stylet 20–22 µm long, cone 61–68% of stylet length and knobs 3–4 µm across. Pharynx well developed, about one-fourth of body length. Hemizonid commonly above secretory-excretory pore about two body annuli long. Secretory-excretory pore between mid-isthmus and end bulb level. Spermatheca rounded to oval and filled with sperm cells. Vulval flaps rounded, prominent. Vulva located at 80–83% of body length from anterior end. Vagina oblique, reaching to half of body width. Tail 21–29 µm long, conical, thin and terminus from spicate to pointed or minutely rounded.

*Molecular characterisation:* Two D2-D3 of 28S, four ITS, four 18S rRNA and four *COI* gene sequences were generated for the first time from this species without intraspecific sequence variations.

*Remarks:* Males were not found. This species is reported for the first time in Belgium and has only been recorded in Poland and Slovakia before in Europe [19,54,55]. Female morphology and morphometrics agree well with the original description [56] and also with descriptions of other populations [19,31]. *Paratylenchus elachistus* can be separated from its closest species, *Paratylenchus minutus*, Lindford in Lindford, Oliveira & Ishii, 1949, by a longer body length (0.23–0.34 mm vs. 0.19–0.31 mm), a more robust and longer stylet (19–25 µm vs. 15–21 µm) and a more slender tail, commonly with spicate to finely rounded tail termini. 

#### 2.1.3. *Paratylenchus goodeyi*


*Females* (Sample BE22; Figure 3, Table 2): Heat relaxed specimens C- to J-shape. Lateral field with four lateral lines. Deirids observed under LM. Cephalic region conical-rounded, submedian lobes not protruding except in two freshly killed specimens where small protrusions were seen under LM. Stylet 47–59 µm long, cone 78–90% of stylet length, stylet guide faintly seen, knobs 3–5 µm across. Pharynx well developed, about one-third of body length. Secretory-excretory pore around median bulb level. Spermatheca oval to elongate, filled with sperm cells. Vulval flaps present. Vulva located at 78–82% of body length from anterior end, in one female a short post-vulva sac observed. Vagina oblique and reaching to two-third of body width. Tail 26–32 µm long, conoid with variable terminus from finely rounded to bluntly rounded and rarely pointed.

*Molecular characterisation:* Three D2-D3 of 28S, one ITS, two 18S rRNA and three *COI* gene sequences were generated for the first time for this species without intraspecific sequence variations.

*Remarks:* Males were not found. Females morphology and morphometrics agree well with former *P. goodeyi* descriptions [18,19,57]. This species was originally described from the Netherlands and has been reported from many European countries, including Belgium. *Paratylenchus goodeyi* is one of the 22 species of the Group 10 of *Paratylenchus* after Ghaderi et al. [23] with stylet length more than 40 µm, four lateral lines and presence of vulval flaps. This species is comparable to other members of the group namely *Paratylenchus ivorensis* Luc & de Guiran, 1962, *Paratylenchus pandatus* (Raski, 1976) Siddiqi, 1989 and *P. straeleni* with females having more or less conical-rounded heads, stylet lengths in the range 40–61 µm (except for *P. pandatus* for which, a stylet length up to 68 µm was rarely reported). However, the vulvae of *P. goodeyi* and *P. straeleni* are located more posteriorly (77–88%) than that of the other two species (70–78%); *P. ivorensis* and *P. goodeyi* have been reported with variable tail termini, while *P. pandatus* and *P. straeleni* have been found usually with finely rounded to sub-acute female tail termini.

#### 2.1.4. *Paratylenchus holdemani*

*Females* (Sample AR3; Figure 4, Table 3): Heat relaxed specimens C- to J-shape. Lateral field with four lateral lines. Deirids not observed. Cephalic region slightly truncated, submedian lobes not protruded. *En face* showing four small submedian lobes, four irregular ridges around oral opening, slit-like lateral amphidial openings. Stylet 19–26 µm long, cone 61–77% of stylet length, knobs 3–4 µm across. Pharynx well developed, about one-fourth of body length. Secretory-excretory pore commonly between mid-isthmus and end bulb level. Spermatheca rounded, filled with sperm cells. Vulval flaps prominent. Vulva located at 81–90% of body length from anterior end. Vagina oblique and reaching to two-third of body width. Tail 20–30 µm long, conoid with regularly finely rounded to sometimes bluntly rounded or digitate terminus.

*Males*: Two males were obtained from Sample AR3 and one from Sample BE20. Their conspecificity with the females was confirmed by identical D2-D3 of 28S rRNA and *COI* gene sequences identified from the AR3 and BE20 males, respectively. The males had an average stylet length of approximately 12 µm and spicule length of 21 µm.

*Molecular characterisation:* Three D2-D3 of 28S, one 18S rRNA and three *COI* gene sequences were generated from the AR3 females, whereas two D2-D3 of 28S, one ITS, one 18S rRNA and four *COI* gene sequences were generated from the BE20 females. No sequence from either population showed any intraspecific variations. The D2-D3 sequences were found to be identical to *P. bukowinensis* sequences that originated from Italy [37] and Belgium [47]; however, morphological data for these populations are not available for comparison and both are considered here as representatives of *P. holdemani*.

*Remarks:* This species has been reported for the first time in Belgium and has only been reported in the Czech Republic in Europe [19]. The morphology and morphometrics of the AR3 population agree well with the original description [58] and with the population from the Czech Republic [19]. Although our D2-D3 sequences were identical to a *P. bukowinensis* sequence (AY780943), the female morphology of this Belgian population is different from *P. bukowinensis* descriptions. 

Most importantly, the average stylet length (22.5 µm) of our population is shorter than for many previously reported *P. bukowinensis* populations. In addition, the cephalic region of *P. bukowinensis* is more rounded than that of *P. holdemani. Paratylenchus holdemani* is comparable to *P. hamatus* and *Paratylenchus baldaccii* Raski, 1975, but is distinguishable from both species by a shorter stylet length of 22.5 ± 2.0 (19–26) µm vs. always above 26 µm. In this study, greater variation in the tail termini was observed in our *P. holdemani* population compared to the other two species.

#### 2.1.5. *Paratylenchus idalimus*

*Females* (Samples BE19 and BE20, two nearby localities; Figure 5 and Figure 6, Table 2): Heat relaxed specimens J- or open C-shape. Lateral field with three lateral lines. Deirids not observed. Cephalic region conical-truncate, submedian lobes well developed and protruding. Stylet 84–93 µm long of which 83–89% is cone, prominent stylet guide, knobs about 4 µm across. Pharynx well developed, occasionally reaching up to half of body length. Secretory-excretory pore around level of stylet knobs which is above median bulb level. Spermatheca small, rounded, usually filled with sperm cells. Vulval lips slightly protruding. Vulval flaps reduced and small, sometimes easily overlooked. Vulva located at 77–79% of body length from anterior end. Vagina oblique, often reaching to two-third of body width. Anus obscure. Tail 22–28 µm long, conoid with subacute to finely rounded terminus.

*Molecular characterisation:* Two identical D2-D3 of 28S and two identical 18S rRNA gene sequences were generated from the BE20 population, whereas one D2-D3 sequence, identical to that of the BE20 sequence, was generated from the BE19 population. These sequences were generated for the first time for this species.

*Remarks:* Males and swollen females were not found. Only one juvenile was recorded from the BE20 population with a stylet length of 42 µm. This is the first time the species has been reported in Europe. Female morphology and morphometrics based on seven females from both populations (three from BE19 and four from BE20) agree well with the description of the slender female by Raski [7] in the USA. This species and *P. colinus* are the only two members of Group 8 of *Paratylenchus* [23]. It differs from *P. colinus* in having a longer stylet (84–93 µm vs. 56–72 µm), more pronounced protrusion of submedian lobes, slightly posterior position of vulva (77–79% vs. 69–78%) and absence vs. presence of cuticular ornamentation in anterior body.

#### 2.1.6. *Paratylenchus microdorus*


*Females* (Sample BE9; Figure 7, Table 3): Body small, heat relaxed specimens open C- to 6-shape. Lateral field with four lateral lines. Deirids not observed. Cephalic region conical-truncate, submedian lobes sometimes slightly protruding. *En face* showing four submedian lobes and slit-like lateral amphidial openings. Stylet 11–15 µm long, cone 45–60% of stylet length. Pharynx about one-fifth of body length. Secretory-excretory pore between mid-isthmus and end bulb level. Spermatheca rounded, empty or filled with sperm cells. Vulval flaps prominent. Vulva located at 79–83% of body length from anterior end. Vagina oblique, reaching to half of the body width. Post-vulval uterine sac not seen. Tail 28–36 µm long, conoid and terminus pointed to subacute to sometimes finely rounded.

*Molecular characterisation:* Three D2-D3 of 28S, three ITS, four 18S rRNA and three *COI* gene sequences were generated without intraspecific variability; the ITS and the *COI* sequences are new for this species. Only 300 bp of the D2-D3 sequences were found to be homologous with four *P. microdorus* sequences from Germany (MF325254–MF325257; 98% similarity; 5 bp difference). The 18S rRNA sequences are 98–99% similar with *P. microdorus* from the Netherlands (AY284632 and AY284633; 8–15 out of 880 bp difference).

*Remarks:* Males were not found. Female morphologies and morphometrics agree well with the original description [59] and other populations [16,31], except for a slightly shorter stylet length (11–15 µm vs. 13–18 µm). Wide variations in the tail termini have been reported for this species [31]. However, for the BE9 population, finely rounded to subacute female tail termini were commonly observed. This species is comparable to *Paratylenchus recisus* Siddiqi, 1996, *Paratylenchus variabilis* Raski, 1975 and *P. veruculatus*, with a female stylet length within 11–17 µm, four lateral lines, presence of vulval flaps, secretory–excretory pore at the posterior part of pharynx and vulva located at 78–87% of body length. However, they differ from each another in having conical-truncate heads with sometimes slightly protruded submedian lobes in *P. microdorus*, broadly rounded to truncated head with central swallow depression in *P. recisus*, rounded to almost hemispherical head in *P. variabilis* and low and broadly rounded head in *P. veruculatus.* Only *P. microdorus* and *P. veruculatus* males have been reported to have weak stylets, while stylets in the males of the other two species are degenerated.

#### 2.1.7. *Paratylenchus nanus*


*Females* (Sample BE11; Figure 8, Table 4): Heat relaxed specimens open C- to J-shape. Lateral field with four lateral lines. Deirids not observed. Cephalic region conical-rounded, in some specimens with sloping sides to rounded end, submedian lobes not protruding under LM. *En face* square shaped, revealing four submedian lobes, four distinct ridges around oral opening, lateral ridges slightly larger than dorso-ventral ridges, and two slit-like lateral amphidial openings. Stylet 27–31 µm long, cone 67–78% of stylet length and knobs 3–5 µm across. Pharynx well developed, about one-fourth of body length. Hemizonid just above secretory-excretory pore, about two body annuli long. Secretory-excretory pore between isthmus and end bulb level. Spermatheca rounded and filled with sperm cells. Vulval flaps present. Vulva located at 82–86% of body length from anterior end. Vagina oblique, reaching up to half of body width. Tail 19–26 µm long, conoid, often more pronounced curvature on dorsal side ending with sub-acute to finely rounded terminus.

*Molecular characterisation:* Seven D2-D3 of 28S, four ITS, four 18S rRNA and seven *COI* gene sequences were generated without any intraspecific sequence variations among four *P. nanus* populations—AR3, BE1, BE11 and BE18. The D2-D3 and the ITS sequences were, respectively, identical to KF242194, KF242197 and KF242267, KF242268 of *P. nanus* from Van den Berg et al. [22]

*Remarks:* Only in the BE11 population was a sufficient number of females recovered to allow morphological and morphometrical data comparisons, which agreed well with the original description [60] and subsequent descriptions of *P. nanus* [19,22,58]. Van den Berg et al. [22] reported two sibling species of *P. nanus* with different genotypes—type A and type B (the latter of which was recently transferred to *P. projectus*) [61]. This correction suggests that the available 28S (MN720102–MN720103) and *COI* (MN734387 and MN734388) sequences of *P. nanus* from South Korea [48] were misidentified as they were found to be identical to the *P. projectus* sequences. *Paratylenchus nanus* is very similar to *P. projectus* and *P. neoamblycephalus*. It differs from *P. projectus* in having a conical-rounded vs. more trapezoid head shape and sperm-filled vs. empty spermathecae. It is differentiated from *P. neoamblycephalus* by more rounded vs. oval spermathecae and a conoid tailwith pronounced curvature on the dorsal side ending with a subacute or finely rounded terminus vs. a conoid tail with subacute terminus or almost acute tip. Furthermore, in our study we also observed that the ridges around the oral opening of the freshly killed specimens protruded more in *P. neoamblycephalus* compared to *P. nanus* when observed under LM.

#### 2.1.8. *Paratylenchus neoamblycephalus*


*Females* (Sample BE10; Figure 9, Table 4): Heat relaxed specimens open C-shape. Lateral field with four lateral lines. Deirids clearly visible on SEM images. Cephalic region truncated-rounded, submedian lobes sometimes very slightly protruding under LM. *En face* showing four rounded to oval submedian lobes, four ridges around oral opening, lateral ridges thicker than dorso-ventral ridges and seen as a protruding bi-lobed structure under LM. Stylet 32–34 µm long, cone 63–73% of stylet length, knobs 4–5 µm across. Pharynx about one-fourth of body length. Secretory-excretory pore between mid-isthmus and end bulb level, with swellings sometimes seen at the duct near the opening in freshly killed specimens. Spermatheca oval and filled with sperm cells. Vulval flaps present. Vulva located at 81–84% of body length from anterior end. Vagina oblique and reaching up to half of body width. Tail 20–28 µm long, conoid and terminating to sub-acute to almost acute tip.

*Males:* Two males were obtained with very thin stylets in freshly killed specimens, which were not visible after fixation, and spicules of 24 µm long. Their conspecificity with the females was confirmed by identical *COI* and D2-D3 sequences.

*Molecular characterisation:* Four D2-D3 of 28S, five ITS, six 18S rRNA and eight *COI* gene sequences were generated without intraspecific sequence variations. The 18S and the *COI* sequences are new for this species. The D2-D3 sequences were found to be identical to KF242189 and KF242190 of an unidentified *Paratylenchus* sp.6 from the USA [22], which is considered here as *P. neoamblycephalus*. However, the D2-D3 sequences were only 89% similar (79 out of 710 bp difference) with MG925221 and 92% similar (43 out of 546 bp difference) with MK506807 of *P. neoamblycephalus* from the USA and Iran, and named here as type A and type B, respectively. Interestingly, we observed 17 ambiguous nucleotide sites in the American *P. neoamblycephalus* type B sequence, which was found to be similar to *P. projectus* (previously *P. nanus* type B; KF242198–KF242201; 98% similarity; 16–20 out of 690 bp difference) after Van den Berg et al. [22,61]. On the other hand, the Iranian *P. neoamblycephalus* sequence [24] was similar to *P. nanus* (KF242194 and KF242197; 95% similarity; 27 out of 575 bp difference) [22]. Furthermore, our ITS sequences were only 74% similar (222 out of 865 bp difference) to MK506794 of *P. neoamblycephalus* type A generated from the same Iranian population.

*Remarks:* This species is reported for the first time in Belgium. Female morphology and morphometrics agree well with the original description from Germany [18] and to subsequent descriptions from Poland [19]. *Paratylenchus neoamblycephalus* is very similar to *P. nanus* and a comparison is provided above.

#### 2.1.9. *Paratylenchus straeleni*

*Females* (Sample BE15; Figure 10, Table 2): Heat relaxed specimens J- to C-shape. Lateral field with four lateral lines. Deirids clearly visible under SEM. Cephalic region conical-rounded to sometimes slightly truncated, submedian lobes not protruded. Stylet straight to slightly curved, 54–59 µm long, cone 76–83% of stylet length, knobs 3–5 µm across. Pharynx roughly one-fourth of body length. Secretory-excretory pore between isthmus and end bulb level. Spermatheca rounded to sometimes slightly ovoid and filled with sperm cells. Vulval flaps distinct. Vulva located at 80–84% of body length from anterior end. Vagina oblique, occasionally reaching to two-third of body width. Tail conical, 31–41 µm long, and terminus sharply pointed to minutely rounded.

*Males*: Two males were recovered without stylets and with spicule lengths of 20 and 22 µm, respectively. Their conspecificity with the females was confirmed by identical *COI* sequences.

*Molecular characterisation:* One D2-D3 of 28S, two ITS (99% similarity; 4 out of 830 bp difference), three identical 18S rRNA and five identical *COI* gene sequences were generated from the BE15 population. From another population (BE11), single D2-D3, ITS and 18S sequences and three identical *COI* sequences were also generated. All the sequences from both populations showed no intraspecific variation, except for the ITS sequences. The 18S sequences were 99% similar (3–5 out of 930 bp difference) with *P. straeleni* from the Netherlands (AY284630 and AY284631). The D2-D3 sequences were 97–99% similar (11–18 out of 700 bp difference) with four *P. straeleni* sequences—i.e., MK506804 from Iran [24], KM875547 from Turkey [42], and KF242235 and KF242236 from the USA [22]. The *COI* sequences were generated for the first time for this species. Remarkably, the Belgian ITS sequences were only 62% similar (295 bp difference) to the Iranian *P. straeleni* sequence (MK506791) of Hesar et al. [24].

*Remarks:* Female morphology and morphometrics agree well with the original description, also from Belgium [62], and subsequent descriptions of globally distributed *P. straeleni* populations [31]. This species is comparable to *P. goodeyi* as described above.

#### 2.1.10. *Paratylenchus veruculatus*

*Females* (Sample BE20; Figure 11, Table 3): Heat relaxed specimens open C-shape to slightly ventrally curved. Lateral field with four lateral lines. Deirids not observed. Cephalic region broadly rounded, submedian lobes not protruding. *En face* rectangular with indistinct submedian lobes, four irregular ridges around oral opening and lateral amphidial openings. Stylet 13–15 µm long, cone 60–65% of stylet length, knobs about 3 µm across. Pharynx roughly one-fourth of total body length. Hemizonid two body annuli long, usually visible just above secretory-excretory pore. Secretory-excretory pore between mid-isthmus and end bulb level. Spermatheca rounded and filled with sperm cells, young females with empty spermatheca also seen. Vulval flaps prominent. Vulva located at 84–90% of body length from anterior end. Vagina oblique and long, reaching up to three-fourth of body width. Tail 14–19 µm long, conoid with often broadly rounded to sometimes finely rounded terminus.

*Molecular characterisation:* Five D2-D3 of 28S (99% similarity; 1–2 out of 720 bp difference) and two identical 18S rRNA and seven *COI* gene (97–100% similarity; 12–13 out of 410 bp difference) sequences were generated for the first time for this species.

*Remarks:* Female morphology and morphometrics agree well with the original description [63] and with other populations [19,31]. This species is comparable to other species with a short stylet such as *P. microdorus*, *P. recisus* and *P. variabilis* (see also above).

#### 2.1.11. *Paratylenchus* sp.2 

*Females* (Sample BE15; Figure 12, Table 4): Heat relaxed specimens open C-shape. Lateral field with four lateral lines. Deirids observed under SEM. Cephalic region conical-rounded, sometimes slightly trapezoid, submedian lobes not protruding under LM. *En face* square-shaped, showing four rounded, poorly separated submedian lobes, four ridges around oral opening, lateral ridges more prominent and larger than dorso-ventral ridges. Stylet 27–31 µm long, cone 64–71% of stylet length, knobs about 4 µm across. Pharynx well developed, about one-fourth of body length. Hemizonid just above secretory-excretory pore, about two body annuli long. Secretory-excretory pore between mid-isthmus and end bulb level. Spermatheca rounded to occasionally slightly ovoid, filled with sperm cells. Vulval flaps prominent. Vulva located at 81–84% of body length from anterior end. Vagina oblique, reaching up to two-third of body width. Tail 23–29 µm long, conoid, slender and terminating with finely rounded tip.

*Molecular characterisation:* Two identical sequences each of D2-D3 of 28S, ITS, 18S rRNA as well as the *COI* gene were generated. The D2-D3 and ITS sequences were found to be, respectively, identical to KF242220 and KF242221 and 99% similar (five out of 750 bp difference) to KF242243 of *Paratylenchus* sp.2, which was identified as a member of the *P. hamatus* species complex [22]. The 18S and *COI* sequences were generated for the first time.

*Remarks:* Males were not found. The female morphology and morphometrics are in agreement with the description of *P. hamatus* [64]. Based on morphology and D2-D3 and ITS sequences, Van den Berg et al. [22] considered *P. hamatus* as a species complex containing several species, including *P. hamatus sensu stricto* collected from the type locality, and *Paratylenchus* sp.1 and *Paratylenchus* sp.2, collected from other places in California. *Paratylenchus* sp.1 is identified as representative of *P. tenuicaudatus. Paratylenchus* sp.2 is morphologically similar with *P. hamatus sensu stricto* but differs based on D2-D3 and ITS sequences [22], and this species appears to be not only present in the USA (California) but also in Belgium and Kyrgyzstan.

#### 2.1.12. *Paratylenchus* sp.BE11

*Females* (Sample BE11; *n* = 3; Figure 13): Body about 0.3 mm long with maximum body width of about 15 µm, heat relaxed specimens open C- to 6-shape. Lateral field with four lateral lines. Deirids not observed. Head broadly rounded, submedian lobes not protruded, cephalic sclerotization strong. Stylet about 15 µm long, cone 60% of stylet length, knobs 3 µm across. Pharynx about one-fourth of body length. Secretory-excretory pore at the level of pharyngeal end bulb or about 70 µm from anterior end. Spermatheca rounded and filled with sperm cells. Vulval flaps small and rounded. Vulva located at 80–82% of body length from anterior end. Tail 25–32 µm, conoid with bluntly rounded tip.

*Molecular characterisation:* Three identical D2-D3 of 28S, one ITS rRNA and two identical *COI* gene sequences were generated.

*Remarks:* No males were found. Female description is based on only three freshly killed specimens, while sufficient specimens are needed for a comprehensive species characterisation. The female morphology is close to *P. variabilis*, *P. veruculatus* and *Paratylenchus vexans* Thorne and Malek, 1986. These four species have more or less broadly rounded heads with non-protruding submedian lobes, stylet lengths in the range of 12–18 µm, four lateral lines, sperm-filled spermathecae, vulval flaps and conoid tails with more or less rounded termini. However, our population appears to have a stronger cephalic sclerotisation and slightly more anteriorly located vulvae (80–82% vs. 80–87%) compared to the other three species. This species is a sister to *P. microdorus* in the D2-D3 tree (96% similarity; 27 out of 740 bp difference), ITS tree (93% similarity; 37 out of 530 bp difference) as well as the COI tree (91% similarity; 36 out of 420 bp difference). It can, however, be readily morphologically distinguished from *P. microdorus* (see above).

#### 2.1.13. *Paratylenchus* sp.D

*Females* (Sample BE20; Figure 14, Table 4): Heat relaxed specimens open C-shape. Lateral field with four lateral lines. Cephalic region conical-rounded to sometimes slightly trapezoid, submedian lobes not protruding under LM. Deirids visible under SEM. *En face* showing four well-separated rounded submedian lobes and four ridges around oral opening. Stylet 26–29 µm long, cone 61–67% of stylet length, knobs about 4 µm across. Pharynx about one-fourth of body length. Hemizonid just above secretory-excretory pore, about two body annuli long. Secretory-excretory pore between mid-isthmus and end bulb level. Spermatheca empty. Vulval flaps prominent, commonly rounded. Vulva located at 82–85% of body length from anterior end. Vagina oblique reaching up to half of body width. Tail 18–26 µm long, conoid with finely rounded to bluntly rounded terminus, sometimes dorsally sinuate.

*Molecular characterisation:* Seven D2-D3 of 28S (99% similarity; one out of 730 bp difference), four ITS, five 18S rRNA and eleven *COI* gene sequences were generated without intraspecific sequence variation.

*Remarks:* Males were not found. The female morphology and morphometrics is close to *P. projectus* and *Paratylenchus neoprojectus* Wu and Hawn, 1975. The cephalic region of females were seen with both rounded to trapezoid shape, secretory-excretory pore located between mid-isthmus to end bulb level, empty spermatheca and tail termini which fit both the above two species. However, the molecular data appears to be different from any available sequences including that of *P. projectus*. A comparative study of this species with type specimens of *P. neoprojectus* and its molecular information should further confirm whether or not this species is *P. neoprojectus*. 

#### 2.1.14. *Paratylenchus* sp.F 

*Females* (Sample BE22; Figure 15, Table 4): Heat relaxed specimen open C- to 6-shape. Lateral field with four lateral lines. Deirids present. Cephalic region conical-truncate, slightly rounded in few specimens, submedian lobes not protruding under LM. *En face* square-shaped, showing four rounded submedian lobes, four ridges around oral opening, dorso-ventral ridges much larger than the lateral ridges, slit like amphidial apertures laterally. Stylet 25–30 µm long, cone 65–70% of stylet length, knobs about 4 µm across. Pharynx about one-fourth of body length. Hemizonid just above secretory-excretory pore, about two body annuli long. Secretory-excretory pore between mid-isthmus and end bulb level. Spermatheca oval to elongated and filled with sperm cells. Vulval flaps rounded to oval and very prominent. Vulva located at 81–84% of body length from anterior end. Vagina oblique, reaching up to two-third of body width. Tail 20–26 µm long, conoid with regularly bluntly rounded terminus.

*Males*: Heat relaxed specimen curved slightly ventrally, about the same body length as females but slightly slender. Cephalic region conoid and rounded. Stylet and pharynx degenerated. Secretory-excretory pore at about one-fifth of body length from anterior end. Spicule arcuate ventrally, about 21.5 µm in length. Gubernaculum 3–5 µm long. Tail conical with finely rounded tip. Conspecificity of males with females was confirmed by identical D2-D3, 18S and ITS sequences.

*Molecular characterisation:* Five D2-D3 of 28S, three ITS, four 18S rRNA and three *COI* gene sequences were generated without intraspecific sequence variations. The D2-D3, 18S and *COI* sequences were found to be identical, respectively, to MN783707, MN783708, MN783668–MN783670 and MN782407–MN782413 of *Paratylenchus* sp.F [47], while the ITS sequences were generated for the first time.

*Remarks:* Specimens belong to the same population as *Paratylenchus* sp.F in Etongwe et al.’s work [47]. Detailed morphological reanalysis revealed very close similarity to *P. nanus*. Nevertheless, the submedian lobes of this species appear to be somewhat more rounded than that of *P. nanus* based on SEM images and the vulval flaps also appear to be more pronounced and rounded compared to that of the latter. However, these characteristics need careful additional observations based on more specimens from both species. All four gene sequences of *Paratylenchus* sp.F were closest to the sequences of *P. elachistus* and phylogenetic analysis revealed their highly supported (PP > 90%) sister relationship. However, this species is morphologically different from *P. elachistus en face*, with rounded vs. poorly differentiated submedian lobes, stylet lengths of 25–30 µm vs. 20–22 µm and bluntly rounded vs. spicate to pointed tail termini.

### 2.2. Phylogenetic and Species Delimitation Analysis

The D2-D3 domains of the 28S rRNA gene alignment (744 bp long) included 128 sequences of 31 *Paratylenchus* species and three outgroup species. Forty-nine new sequences were included in this analysis. The Bayesian 50% majority rule consensus tree inferred from the analysis of the D2-D3 alignment contained three highly supported major clades and a weakly supported one (Figure 16, PP < 70%). The molecular species delimitation based on the generalized mixed-yule coalescent (GMYC) and Poisson tree process (bPTP) methods revealed 66 and 63 putative species, respectively, a result that is largely congruent with former species delineations. However, *P. projectus*, *P. straeleni*, *P. minor* and *P. shenzhenensis* were further divided into 6, 5 (four according to bPTP), 2 and 2 separate lineages, respectively.

The ITS rRNA gene alignment (995 bp long) included 99 sequences of 37 *Paratylenchus* species and three outgroup species. Thirty-six new sequences were included in this analysis. The Bayesian 50% majority rule consensus tree inferred from the analysis of the ITS alignment contained four highly supported major clades (Figure 17). Results of molecular species delimitation showed a high discrepancy between the models used—i.e., 48 putative species based on GMYC vs. 56 species based on bPTP. Additionally, molecular species delimitation based on the GMYC and bPTP methods did not correspond to species demarcation based on morphology and clade support; for example, virtually all individual sequences of *P. chongqingensis* and *P. shenzhenensis* were delineated as separate species.

The 18S rRNA gene alignment (899 bp long) included 88 sequences of 31 *Paratylenchus* species and two outgroup species. Fifty-four new sequences were obtained for this study. The Bayesian 50% majority rule consensus tree inferred from the analysis of the partial 18S sequence alignment contained four highly supported major clades (Figure 18). Molecular species delimitation failed to delimit well established species—for example *P. goodeyi, P. veruculatus, P. nanus* and *P. neoamblycephalus* were identified as belonging to the same species. Furthermore, both models provided highly varied results (14 putative species according to GMYC vs. 26 according to bPTP), reducing the confidence in said results.

The *COI* gene alignment (745 bp long) included 130 sequences of 31 *Paratylenchus* species and three outgroup species. Seventy-one new sequences were included in this analysis. The Bayesian 50% majority rule consensus tree inferred from the analysis of the *COI* sequence alignment contained four moderate (Figure 19, PP = 70–90%) or highly supported major clades. Both employed species delineation methods, GMYC and bPTP, provided exactly the same 54 putative species delineations. These results were largely consistent with those obtained using other methods. However, *P. enigmaticus*, *P. microdorus* and *P. veruculatus* were subdivided into different species despite these sequences originating from the same population and their corresponding D2-D3 sequences being similar. *Paratylenchus straeleni* was appointed as a species complex with nine putative species. Statistical parsimony networks showing the phylogenetic relationships between different isolates of *P. straeleni* and *P*. *enigmaticus* based on *COI* sequences are given in Figure 19B,C. The maximum variation of sequences for *P. straeleni* was found to be 9.1%.

Taking both morphological and molecular evidence together, we have been able to reassign a total of 49 *Paratylenchus* sequences, including 18 D2-D3 of 28S, 3 ITS, 3 18S rRNA and 25 *COI* gene sequences, to their appropriate species (Table 5). However, we cannot exclude that in future, the identification of *Paratylenchus* species made in this study may be improved in light of new datasets.

## 3. Discussion

The genus *Paratylenchus sensu lato*, with 124 valid species, is an important plant-parasitic group consisting of several commonly occurring and economically important species such as *P. bukowinensis, P. dianthus, P. hamatus, P. nanus, P. neoamblycephalus* and *P. projectus*, which are difficult to separate solely based on morphology [14,23,25,31,32,33,34,35,36]. Female morphological traits are the most commonly used features for the identification of *Paratylenchus* populations, with the relative lengths of stylet cones and the positions of the secretory-excretory pores and vulvae as the most informative traits [31,67], while several ratios such as a, c and c’ show high intraspecific variation. Given the limited species-specific female traits, some characteristics of males and juveniles—such as the presence or absence of stylet and male spicule length—may also be used to supplement the available data. However, care must be taken—for example, the occasional observance of a thin stylet in freshly killed juveniles or males that was invisible once the specimens were fixed highlights the importance of reporting this characteristic from both freshly killed and fixed specimens. Further complicating *Paratylenchus* taxonomy is the presence of mixture of species within one locality and sample [31]—an observation which calls for precaution concerning the conspecificity of several life stages. Indeed, the presence of multiple species in a soil sample was amply illustrated in our study. Seventy five percent of our investigated soil samples contained multiple species, with up to five different *Paratylenchus* species present in the same sample. This is, to the best of our knowledge, one of the highest numbers of species of one plant-parasitic nematode genus present in a single soil sample. More suitable morphological characters such as ridges around the oral opening or distinct to fused submedian lobes in face view also appear to be usefully informative but were only clearly revealed in our study with supporting evidence from SEM; additionally, the small vulval flaps in *P. aculentus* confirmed in this study have often been overlooked in previous studies under LM. Scanning electron microscopy is known to be important in nematode taxonomy [68,69,70], and this is especially true for the genus *Paratylenchus* as demonstrated in this study. 

Nevertheless, even if all existing morphological tools are carefully employed, it remains impossible for all *Paratylenchus* species to be morphologically delineated, owing to the existence of cryptic species such as *P. aquaticus* [22]. The extensive use of new molecular data in the current study has demonstrated a remarkable molecular diversity in *Paratylenchus*, with several additional cryptic species being potentially present. The most obvious example is *P. straeleni*, which comprises 9, 5 and 4 putative species according to *COI-* (both GMYC and bPTP), D2-D3 (GMYC) and D2-D3 (bPTP)-based molecular species delimitation methods, respectively. It is noteworthy that the *P. straeleni COI* sequences have clearly clustered according to geographical location, as revealed by the COI haplotype network. The problems of morphologically delineating the *Paratylenchus* species have been further demonstrated in our study by the difficulties experienced in distinguishing between *Paratylenchus* sp.2, *Paratylenchus* sp.D and *Paratylenchus* sp.F, which were found to be very similar to *P. hamatus, P. projectus*/*P. neoprojectus* and *P. nanus*, respectively. A formal description with an appropriate diagnosis can only be developed for these putative new species following detailed observations of additional specimens and a thorough comparison with type materials of the known species. 

Taken together, it is abundantly clear that molecular data are essential in advancing *Paratylenchus* taxonomy. Unfortunately, the several sequences published for *Paratylenchus* have serious limitations. One such issue is that the majority of the available D2-D3, 18S and ITS rRNA sequences have either not been linked to morphological data or have been associated with poor morphological data, thereby rendering them unreliable for use in identification purposes. For example, sequences of *P. aculentus*, *P. leptos, P. microdorus, P. neoamblycephalus*, etc., are not currently linked to reliable and clear morphological data and any subsequent identification based on these sequences may, therefore, lead to the deposition of further sequences under incorrect names [51]. An additional problem identified with the currently available 18S sequences, which are often relatively short (700–800 bp), is that several *Paratylenchus* species were detected with almost identical sequences. It is clear that to render these conserved sequences useful, complete or nearly complete lengths of the 18S rRNA gene (1600–1800 bp) will be required to allow species delimitation [71,72,73,74].

In the present study, we have also applied DNA-based species delimitation approaches to infer putative species boundaries on a given phylogenetic input, based on two different models [75,76] and four gene fragments (D2-D3 of 28S, ITS, 18S rRNA and *COI*). These coalescence-based species delimitation methods are rapidly gaining popularity in studies on closely related species that are difficult to distinguish based on phenotypic features, and have been applied to various eukaryotic groups [77,78]. However, despite plant-parasitic nematodes being a morphologically minimalistic group par excellence, such methods have been rarely applied to this group; nevertheless, they appear to be largely congruent with traditional methods [79,80,81]. Conversely, we have observed a remarkable discrepancy among the genes used, showing a poor link between DNA species delimitation and other methods, including a discrepancy between the employed models. The ITS and 18S rRNA genes gave, respectively, a likely overestimation and underestimation of the number of putative species, while for *COI* and D2-D3 of 28S rRNA genes, we observed, to a certain extent, an agreement with traditional methods, albeit with a likely overestimation of the number of species in several cases. This was not unexpected, as it has been exemplified by several studies that methods of species delimitation based on the coalescent model tend to overestimate phylogenetic lineages [52,77,82]. Both approaches (bPTP and GMYC) are similar in the fact that they identify significant changes in the pace of branching events on the tree. However, GMYC uses time to identify branching rate transition points, whereas, in contrast, bPTP directly uses the number of substitutions. Based on real and simulated data, both methods yield, in general, similar results [76,83]. This is the case for our COI-based output (identical results) and the D2-D3-based output (two differences, bPTP being more conserved). If differences have been observed, bPTP usually yields a more conservative delimitation than GMYC [76,80,83]. This is contrary to our unexpected ITS and 18S results and reduces the trust in the latter. Counterintuitively, the mutation rate of a chosen marker does not have a direct influence on its effectiveness to detect species. Mitochondrial markers reveal clearer discontinuities between interspecific divergence and intraspecific variation because of their faster coalescence within species lineages compared with nuclear loci, not necessarily because of their higher mutation rates [84,85]. The discrepancy between ITS and other delimitation methods in this study agrees with previous observations pointing to an unclear transition between species-level and population-level genetic distance for ITS [78]. Furthermore, it has been indicated that species delimitation based on single gene trees has serious limitations due to gene tree-species tree incongruence—confusions caused by processes including incomplete lineage sorting, trans-species polymorphism, hybridisation and introgression [78]. Multilocus approaches provide a posteriori double-check for contamination, sequencing errors or mitochondria-specific pitfalls [86]—for example, the high *COI* gene sequence variations within *P*. *enigmaticus*, *P. microdorus* and *P. veruculatus* observed in this study, despite these sequences originating from the same population. Although both nuclear and mitochondrial sequences were provided consistently from the same morphologically vouchered individuals, this study was restricted to the use of only single-locus data since only a limited number of other *Paratylenchus* individuals (and plant-parasitic nematodes in general) are linked to the same two genes. A further rigorous acquisition of both D2-D3 of 28S and *COI* gene sequences, which appear most promising for species delimitation in plant-parasitic nematodes (see [74]), will allow for more substantiated coalescence-based, multilocus species delimitation in plant-parasitic nematodes. Nevertheless, based on all obtained evidence, our findings support the proposition of Puillandre et al. [87], Padial et al. [88] and Qing et al. [80], that DNA-based species delimitation methods are important tools for the exploration of species delineation in diverse groups, but that identification of any new putative species will require further corroboration by an integrative taxonomic approach.

## 4. Materials and Methods

### 4.1. Nematode Populations

Nematode samples used in this study were collected from various localities (Table 1). Bulk soil samples of about 500 mL from 15–20 cm depths were collected from twelve locations in Belgium using a shovel. They were subsequently stored at 4 °C until nematode extraction. Nematodes were extracted from soil using a modified Baermann’s method [89] or a rapid centrifugal flotation method [90]. Nematode extracts were observed under a stereo microscope. *Paratylenchus* populations were picked out in an embryo glass dish and stored in tap water at 4 °C for further analysis.

### 4.2. Morphological Study

Morphological study of nematodes was carried out using both heat relaxed and fixed specimens mounted on temporary and permanent slides, respectively. For preparation of a temporary mount of a nematode, a Cryo-Pro label (VWR International) was cut into two halves and stuck at the centre of a glass slide creating a small parallel gap between them. A single nematode was then transferred in a drop of distilled water to the glass slide in the centre of the gap. The nematode was then heat relaxed by passing over a flame a few times and covered with a glass coverslip. The specimen was then examined, photographed and measured using an Olympus BX51 DIC Microscope (Olympus Optical, Tokyo, Japan), equipped with an Olympus C5060Wz camera [91]. After recording morphological data, the specimen was recovered from the slide by adding a few drops of water from one end of the gap and collecting the nematode that was flushed out on the other end of the gap. The recovered specimens were subsequently used to extract genomic DNA as described in the next section.

A small nematode suspension of the remaining nematodes was heated in an embryo glass dish with a few drops of Trump’s fixative ((2% paraformaldehyde, 2.5% glutaraldehyde in 0.1 M Sorenson buffer (sodium phosphate buffer at pH = 7.5)) in a microwave (700 Watts) for 3–4 sec and leaving it at room temperature for 1 h and at 4 °C for 24 h and followed by gradually transferring to anhydrous glycerine, as described in Singh et al. [92]. The fixed specimens were then mounted in glycerine on glass slides and were studied as above using the camera-equipped microscope. Species identification was carried out both at Nematology Research Unit of Ghent University and National Plant Protection Organization, Wageningen, the Netherlands.

For scanning electron microscopy, specimens fixed in Trump’s fixative were washed in 0.1 M phosphate buffer (pH = 7.5) and dehydrated in a graded series of ethanol solutions, critical point-dried with liquid CO_2_, mounted on stubs with carbon tabs (double conductive tapes), coated with gold of 25 nm, and photographed with a JSM-840 EM (JEOL) at 12 kV [92].

### 4.3. Extraction of DNA, PCR and Sequencing

Genomic DNA was extracted from individual heat relaxed nematode specimen, which had been morphologically vouchered. The cuticle of the specimen was punctured using a fine entomological pin mounted on a thin bamboo stick, which was also used as nematode picking tool and the nematode was subsequently transferred to a PCR tube with 20 µL of worm lysis buffer (50 mM KCl, 10 mM Tris at pH = 8.3, 2.5 mM MgCl_2_, 0.45% NP 40 (Tergitol Sigma), 0.45% Tween 20) and incubated at −20 °C (at least 10 min). This was followed by adding 1 µL proteinase K (1.2 mg/mL), incubation at 65 °C (1 h) and 95 °C (10 min) and ending by centrifuging the mixture at 14,000 rpm for 1 min [92]. Genomic DNA from a single nematode was used to amplify four DNA fragments—D2-D3 of 28S, partial ITS and partial 18S rRNA gene and partial *COI* gene of mtDNA. PCR and sequencing were completed in two laboratories: Nematology Research Unit, Gent University, Belgium and Nematology lab, Plant Pest Diagnostic Center, CDFA, Sacramento, California, USA. For PCR amplifications of the D2-D3 of 28S, ITS and 18S rRNA gene sequences, the primer pairs D2A: 5′-ACA AGT ACC GTG AGG GAA AGT TG-3′/D3B: 5′-TCC TCG GAA GGA ACC AGC TAC TA-3′ [93], Vrain2F: 5′-CTT TGT ACA CAC CGC CCG TCG CT-3′/Vrain2R: 5′-TTT CAC TCG CCG TTA CTA AGG GAA TC-3′ [94] or TW81: 5′-GTT TCC GTA GGT GAA CCT GC-3′/AB28: 5′-ATA TGC TTA AGT TCA GCG GGT-3′ [95], and SSU18A: 5′-AAA GAT TAA GCC ATG CAT G-3′/SSU26R: 5′-CAT TCT TGG CAA ATG CTT TCG-3′ [96] were used, respectively, with thermal profiles described by Singh et al. [97] and Tahna Maafi et al. [98]. Partial *COI* gene was amplified using the primer pairs JB3: 5′-TTT TTT GGG CAT CCT GAG GTT TAT-3′/JB4.5: 5′-TTT TTT GGG CAT CCT GAG GTT TAT-3′ according to Bowles et al. [99] or COI-F5: 5′-AAT WTW GGT GTT GGA ACT TCT TGA AC-3′/COI-R9: 5′-CTT AAA ACA TAA TGR AAA TGW GCW ACW ACA TAA TAA GTA TC-3′ according to Powers et al. [100]. The PCR products were purified [101] and sent to Macrogen [102] and Genewiz [103] for sequencing. New sequences were assembled using Geneious Prime 2020.0.5 and deposited to the GenBank under the accession numbers given in Table 1.

### 4.4. Phylogenetic and Species Delimitation Analysis

The new sequences for each gene (D2-D3 of 28S, ITS, 18S rRNA and *COI*) were aligned using Clustal X 1.83 [104] with their corresponding published gene sequences [22,24,32,33,34,35,36,37,38,39,42,43,44,45,46,47,48,49,50]. Outgroup taxa for each dataset were chosen based on previously published data [105]. Sequence datasets were analysed with Bayesian inference (BI) using MrBayes 3.1.2 [106] under the GTR + I + G model. BI analysis was initiated with a random starting tree and was run with four chains for 1.0 × 10^6^ generations for 18S and ITS rRNA gene alignments, 5.0 × 10^6^ generations for D2-D3 of 28S rRNA gene alignment and 9.0 × 10^6^ generations for *COI* gene alignment. The Markov chains were sampled at intervals of 100 generations. Two runs were performed for each analysis. The log-likelihood values of the sample points stabilised after approximately 1,000 generations. After discarding burn-in samples and evaluating convergence, the remaining samples were retained for further analysis. The topologies were used to generate a 50% majority rule consensus tree. Posterior probabilities (PPs) are given on appropriate clades. Sequence analyses of alignments were performed with PAUP^∗^ 4b10 [107]. Pairwise divergences between taxa were computed as absolute distance values and as percentage mean distance values based on whole alignment with adjustments for missing data. 

The *COI* gene alignments for *P. straeleni* and *P. enigmaticus* were used to construct phylogenetic network estimation using statistical parsimony, as implemented in POPART software [108].

Species delimitation of *Paratylenchus* in this study was undertaken using an integrated approach that considered morphological and morphometric evaluations combined with molecular-based phylogenetic inference (tree-based methods) and coalescent-based molecular species-delimitation methods. Putative species boundaries on a given phylogenetic input tree were inferred using a Bayesian implementation of the Poisson tree processes (bPTP) method [76] and using the generalized mixed-yule coalescent (GMYC) method [75]; see Qing et al. [80] for more details. Ultrametric trees were constructed using BEAST v1.10.4 [109] based on D2-D3, ITS, 18S and *COI* sequences, respectively. Default prior distributions were used and analyses were run for 1 × 10^7^ generations, saving trees every 1 × 10^3^ generations. The final trees were produced after removing 2,000 samples (20%) as burn-ins, and the maximum clade credibility tree was calculated using TreeAnnotator 1.10.4 [109]. Finally, for the bPTP method, an unrooted Bayesian 50% majority-rule consensus tree, containing only ingroups and unique haplotypes, was uploaded on the online server [110] and 1 × 10^5^ Markov chain Monte Carlo (MCMC) generations were performed. The same tree was also uploaded on the GMYC web server [111] using the single threshold method. 

## 5. Conclusions

An integrative approach by linking DNA sequences and morphological characters represents the best way to move nematode taxonomy forward. Creating this link involves the rigorous generation of multiple DNA sequences from individual morphologically vouchered nematode specimens, which, in the current study, resulted in the first molecular characterisations for five species, the first *COI* sequences for eight species and, most importantly, the reassignments of 18 D2-D3 of 28S, 3 ITS, 3 18S rRNA and 25 *COI* gene sequences, which had been unidentified or misidentified.

This study showed that *Paratylenchus* is a case in point, representing an incredibly diverse yet morphologically minimalistic plant-parasitic genus. Our recommendations for future protocol in *Paratylenchus* taxonomy, which are also valid for integrative nematode taxonomy, are: (1) to include SEM in new descriptions or re-descriptions; (2) to use juvenile and male traits after their conspecificity is irrefutably proven using molecular data; (3) to unequivocally link elaborate morphological data with both nuclear D2-D3 of 28S rRNA and mitochondrial *COI* gene sequences; (4) to employ caution when performing molecular identification using partial 18S rRNA gene fragments only; (5) to make use of the promising molecular species delineation methods to establish species boundaries, but base this on multilocus data and merely use it as one of the elements of integrative taxonomy.

## Figures and Tables

**Figure 1 plants-10-00408-f001:**
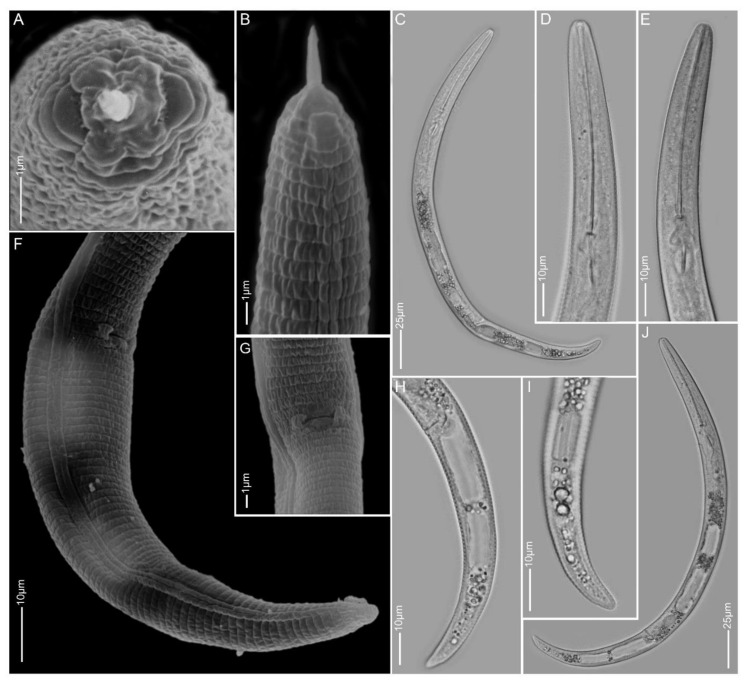
Light and scanning electron microscopy images of *Paratylenchus aculentus* females: (**A**) face view; (**B**,**D,E**) anterior region; (**C,J**) total body; (**G**) vulva region; (**F**,**H**,**I**) tail region.

**Figure 2 plants-10-00408-f002:**
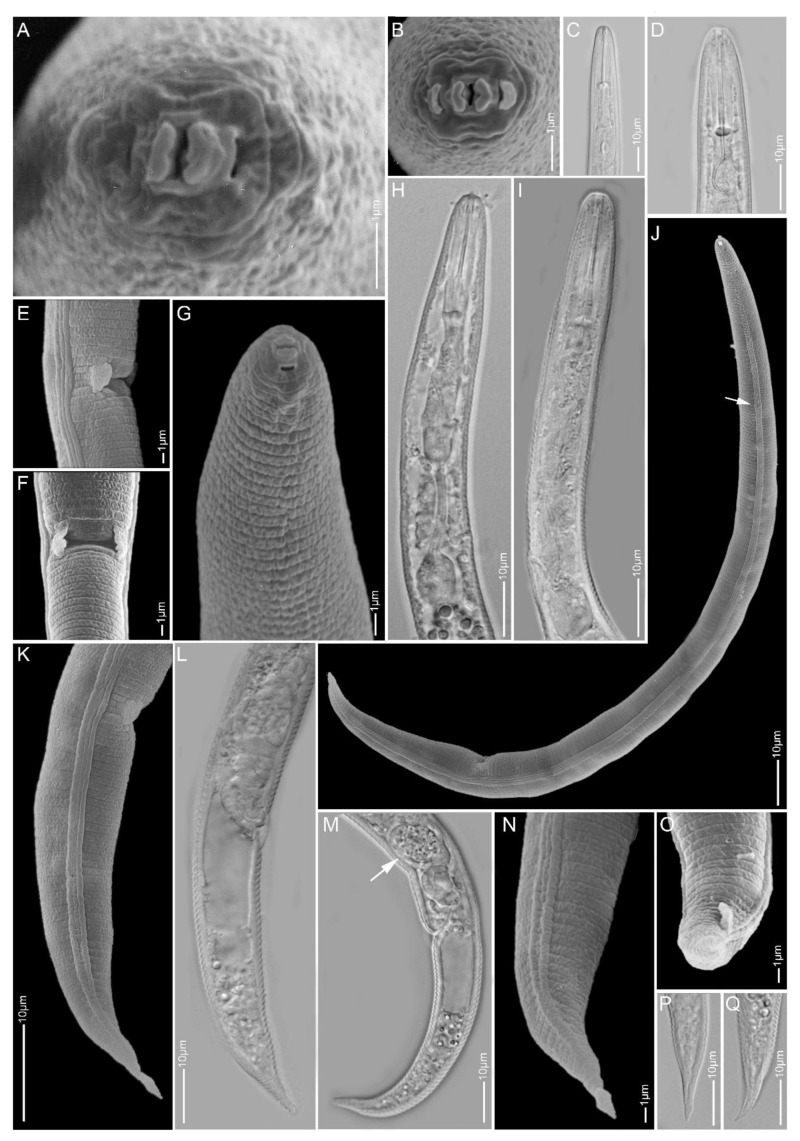
Light and scanning electron microscopy images of *Paratylenchus elachistus* females: (**A**,**B**) face view; (**C**,**D**,**G**–**I**) anterior region; (**E**,**F**) vulva region; (**J**) total body; (**K**–**Q**) tail region; arrows pointed to deirid in (**J**) and spermatheca in (**M**).

**Figure 3 plants-10-00408-f003:**
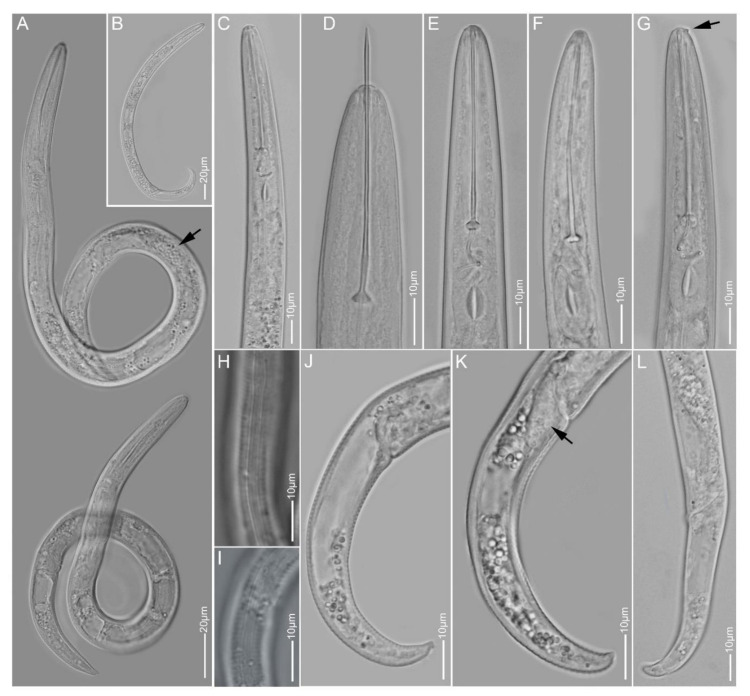
Light microscopy images of *Paratylenchus goodeyi* females: (**A**,**B**) total body; (**C**–**G**) anterior region; (**H**,**I**) lateral field; (**J**–**L**) tail region; arrows pointed to spermatheca in A, protruding submedian lobe in G and post-vulva sac in K.

**Figure 4 plants-10-00408-f004:**
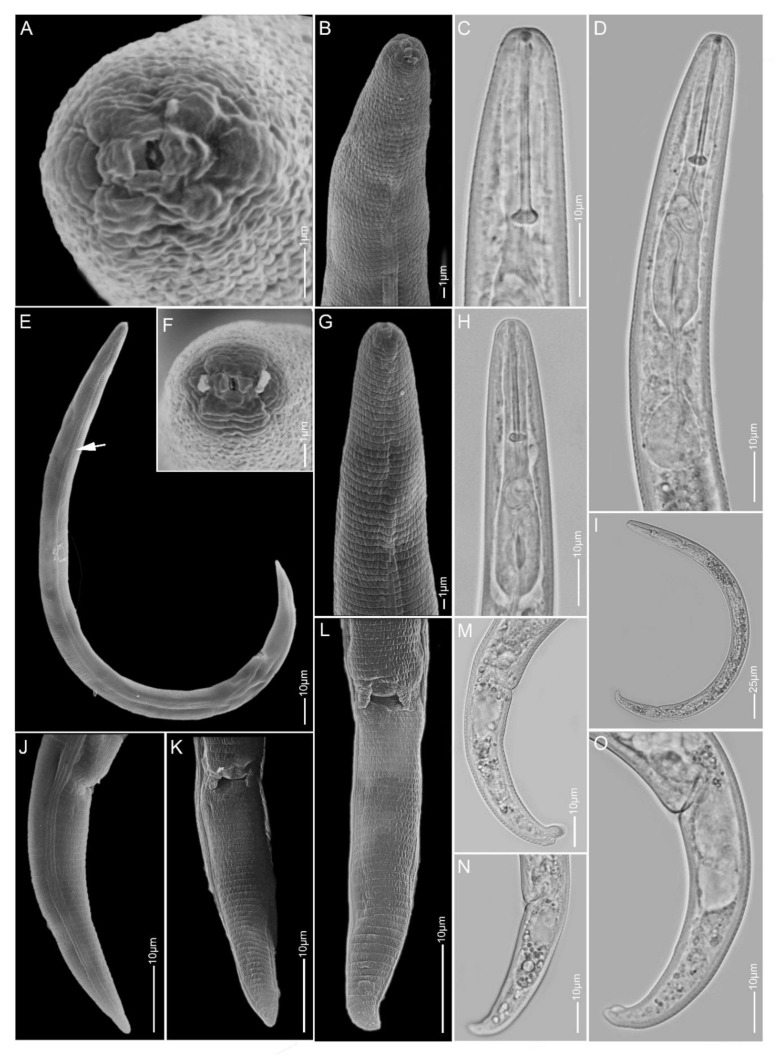
Light and scanning electron microscopy images of *Paratylenchus holdemani* females: (**A**,**F**) face view; (**B**–**D**,**G**,**H**) anterior region; (**E**,**I**) total body; (**J**–**O**) tail region; arrow pointed to deirid in (**E**).

**Figure 5 plants-10-00408-f005:**
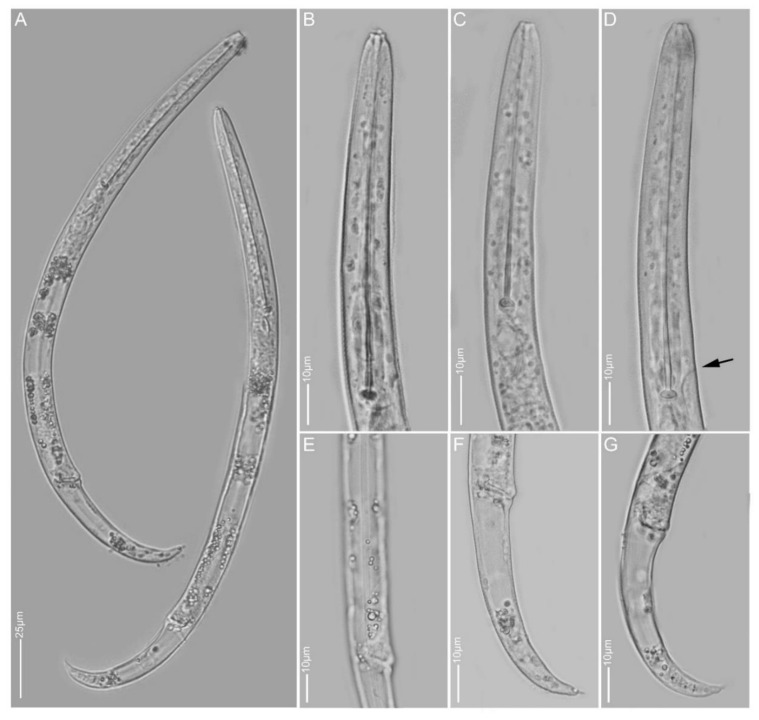
Light microscopy images of *Paratylenchus idalimus* females from sample BE19: (**A**) total body; (**B**–**D**) anterior region; (**E**–**G**) lateral field and tail region; arrow pointed to secretory–excretory pore in D.

**Figure 6 plants-10-00408-f006:**
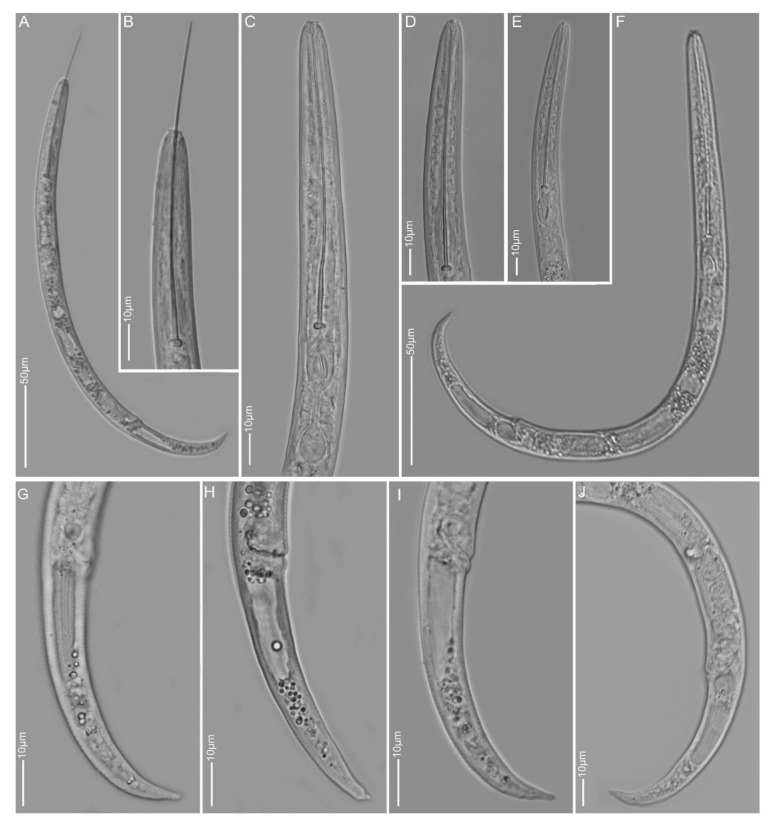
Light microscopy images of *Paratylenchus idalimus* females from sample BE20: (**A**,**F**) total body; (**B**–**E**) anterior region; (**G**–**J**) tail region.

**Figure 7 plants-10-00408-f007:**
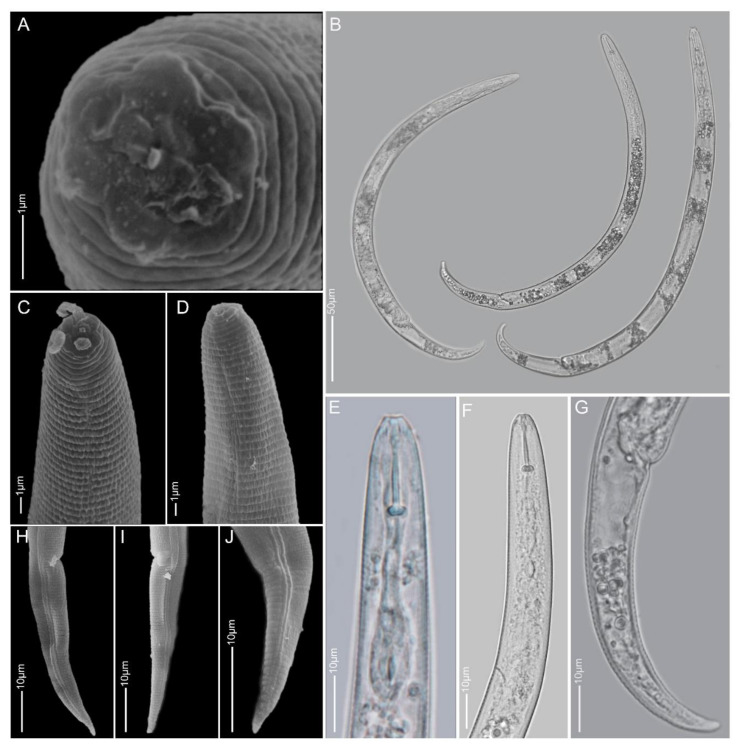
Light and scanning electron microscopy images of *Paratylenchus microdorus* females: (**A**) face view; (**B**) total body; (**C**–**F**) anterior region; (**G**–**J**) tail region.

**Figure 8 plants-10-00408-f008:**
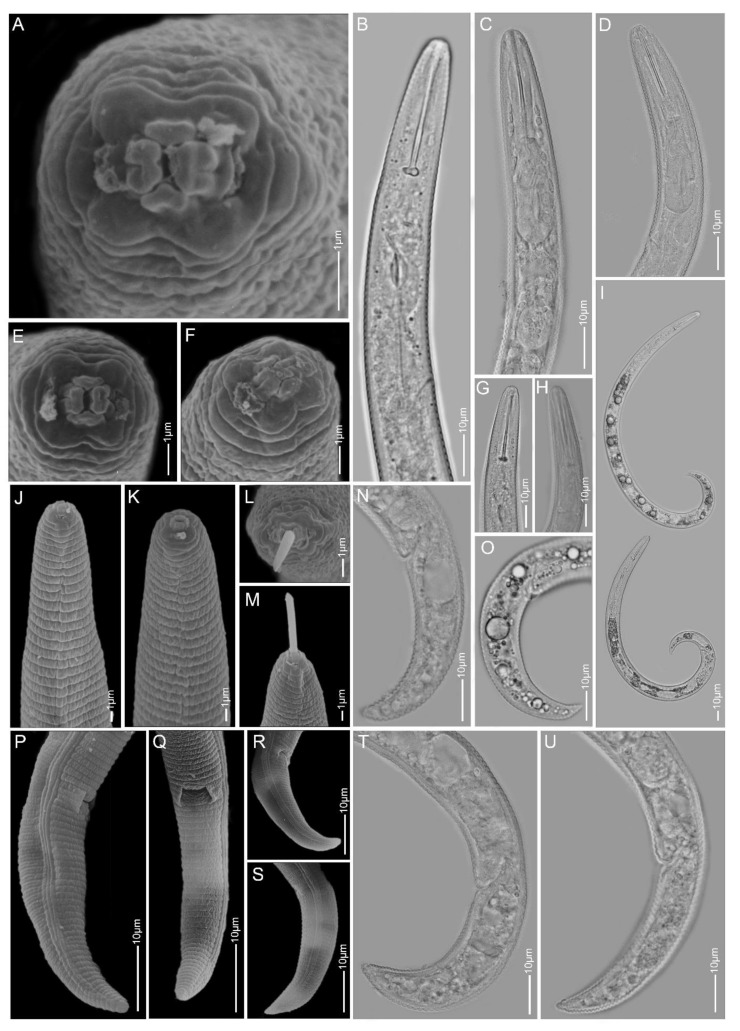
Light and scanning electron microscopy images of *Paratylenchus nanus* females: (**A**,**E**,**F**,**L**) face view; (**B**–**D**,**G**,**H**,**J**,**K**,**M**) anterior region; (**I**) total body; (**N**,**O**,**P**–**U**) tail region.

**Figure 9 plants-10-00408-f009:**
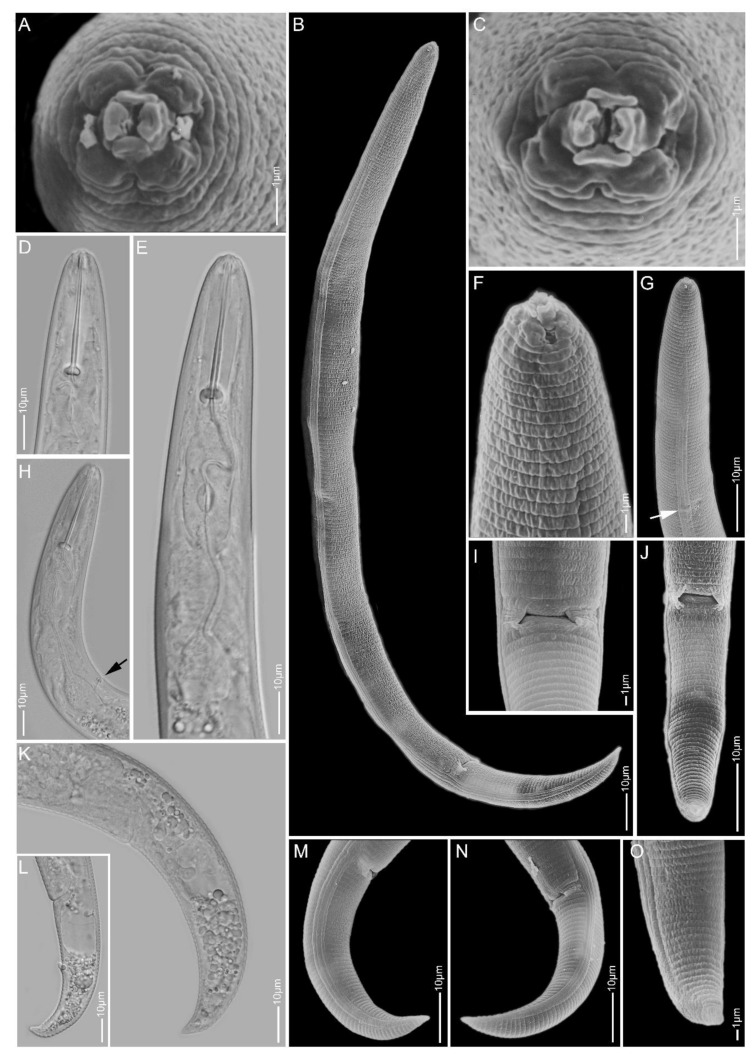
Light and scanning electron microscopy images of *Paratylenchus neoamblycephalus* females: (**A**,**C**) face view; (**B**) total body; (**D**–**H**) anterior region; (**I**–**O**) tail region; arrows pointed to secretory–excretory pore in (**H**) and deirid in (**G**).

**Figure 10 plants-10-00408-f010:**
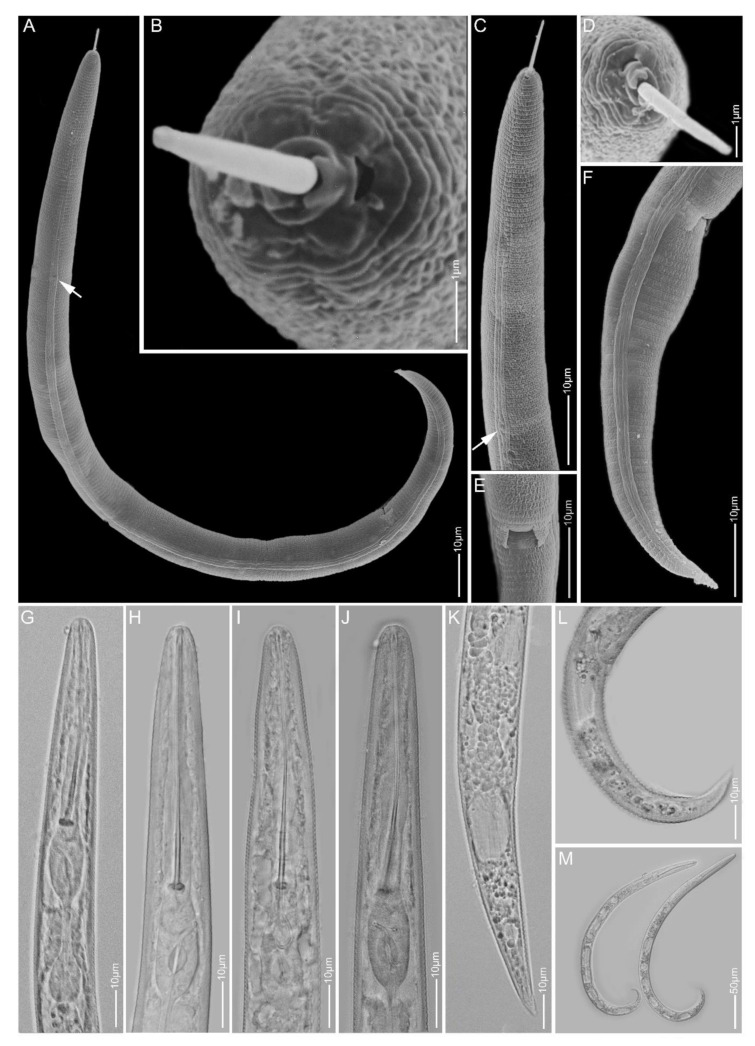
Light and scanning electron microscopy images of *Paratylenchus straeleni* females: (**A**) whole body; (**B**,**D**) face view; (**C**,**G**–**J**) anterior region; (**E**) vulva region; (**F**,**K**,**L**) tail region; (**M**) total body; arrows pointed to deirids in (**A**,**C**).

**Figure 11 plants-10-00408-f011:**
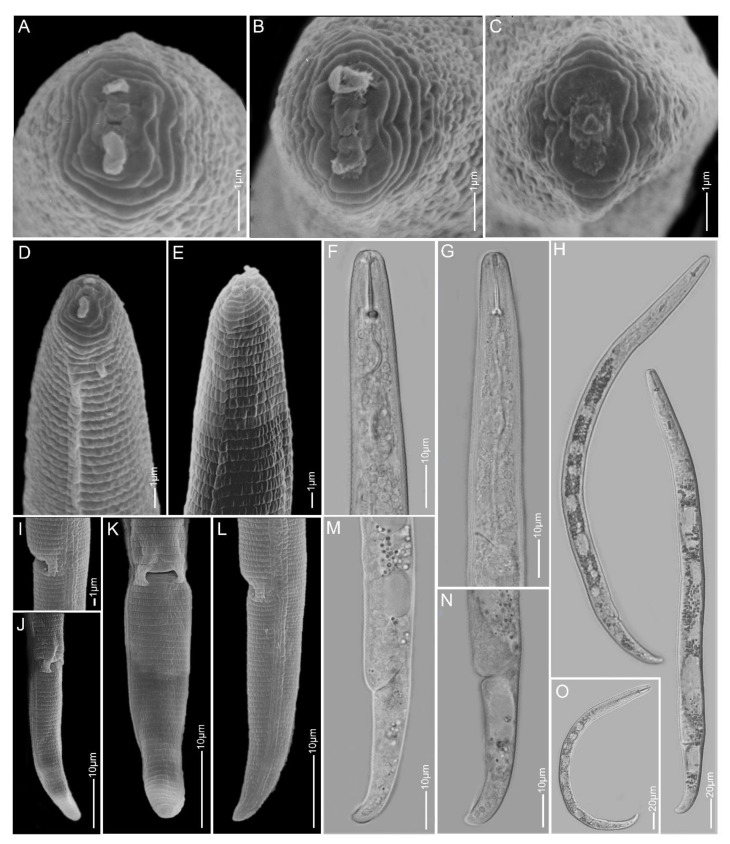
Light and scanning electron microscopy images of *Paratylenchus veruculatus* females: (**A**–**C**) face view; (**D**–**G**) anterior region; (**H**,**O**) total body; (**I**–**N**) lateral field and tail region.

**Figure 12 plants-10-00408-f012:**
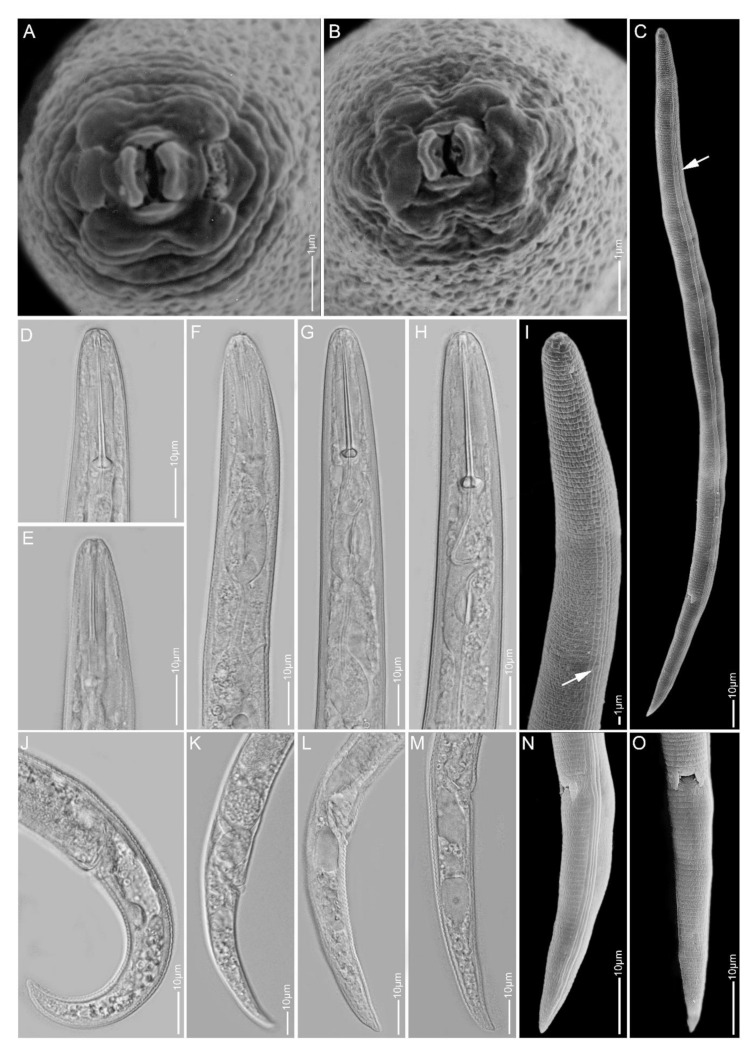
Light and scanning electron microscopy images of *Paratylenchus* sp.2 females: (**A**,**B**) face view; (**C**) total body; (**D**–**I**) anterior region; (**J**–**O**) tail region; arrows pointed at deirids in (**C**,**I**).

**Figure 13 plants-10-00408-f013:**
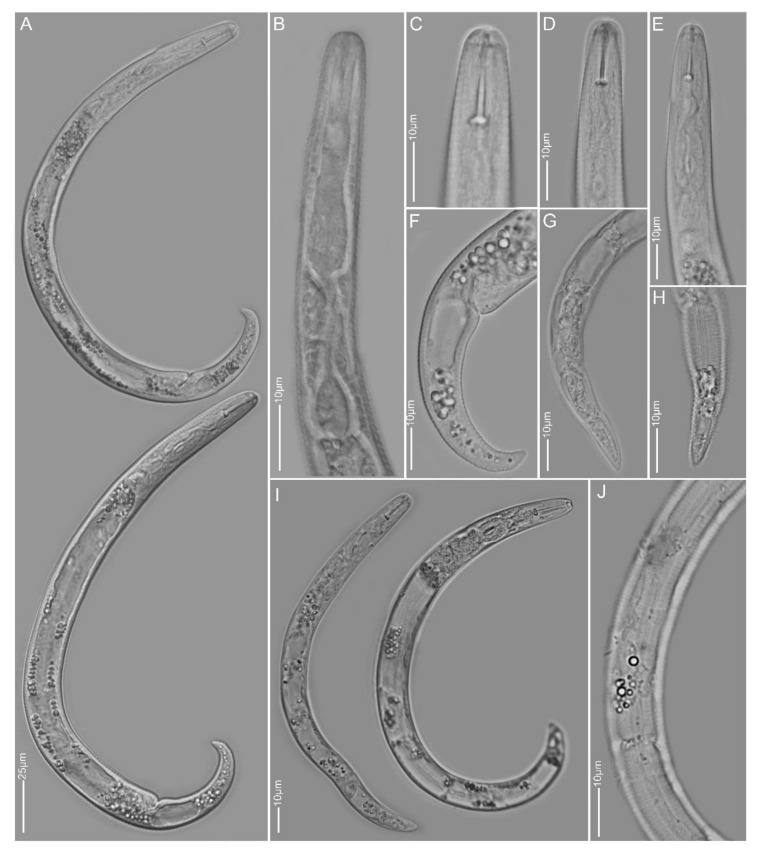
Light microscopy images of *Paratylenchus* sp.BE11 females: (**A**) total body; (**B**–**E**) anterior region; (**F**–**H**) tail region; (**I**) total body; (**J**) lateral field.

**Figure 14 plants-10-00408-f014:**
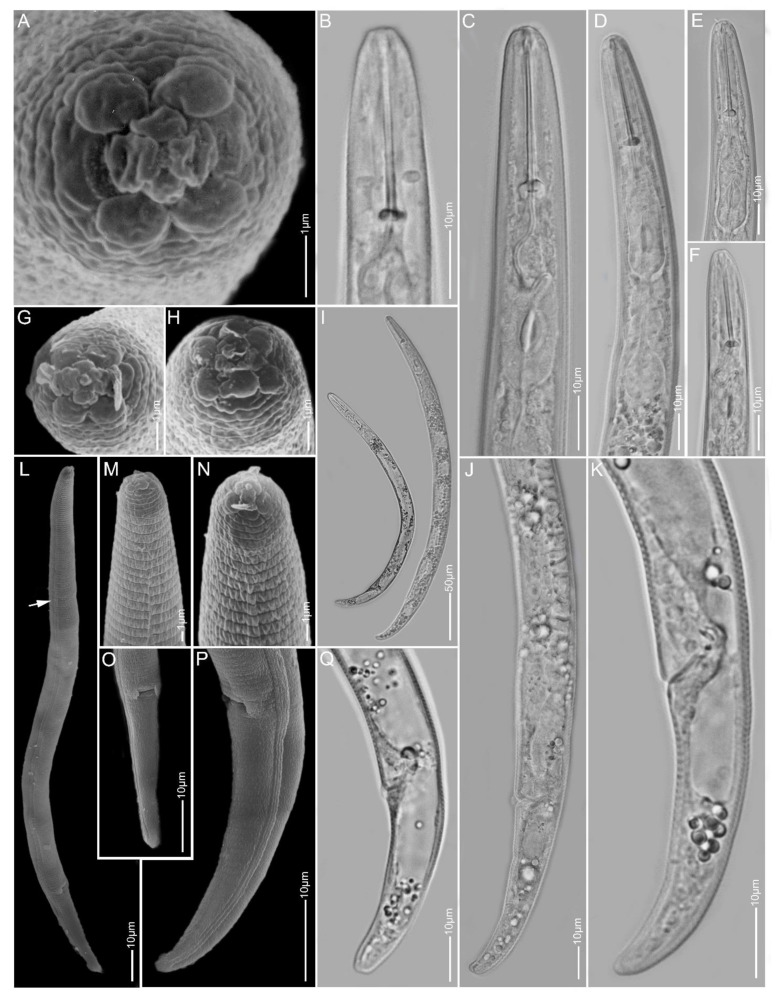
Light and scanning electron microscopy images of *Paratylenchus* sp.D females; (**A**,**G**,**H**) face view; (**B**–**F**,**M**,**N**) anterior region; (**I**,**L**) total body; (**J**,**K**,**O**–**Q**) tail region; arrow pointed to deirid in L.

**Figure 15 plants-10-00408-f015:**
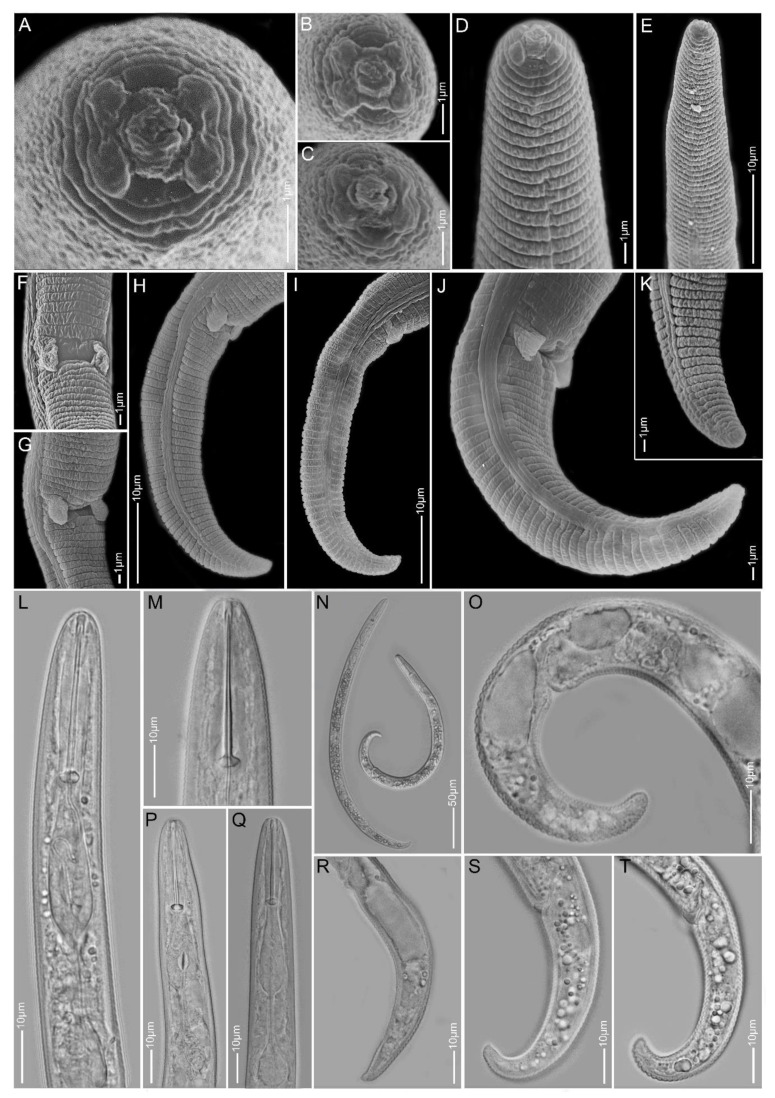
Light and scanning electron microscopy images of *Paratylenchus* sp.F females: (**A**–**C**) face view; (**D**,**E**,**L**,**M**,**P**,**Q**) anterior region; (**F**,**G**) vulva region; (**H**–**K**,**O**,**R**–**T**) tail region; (**N**) total body.

**Figure 16 plants-10-00408-f016:**
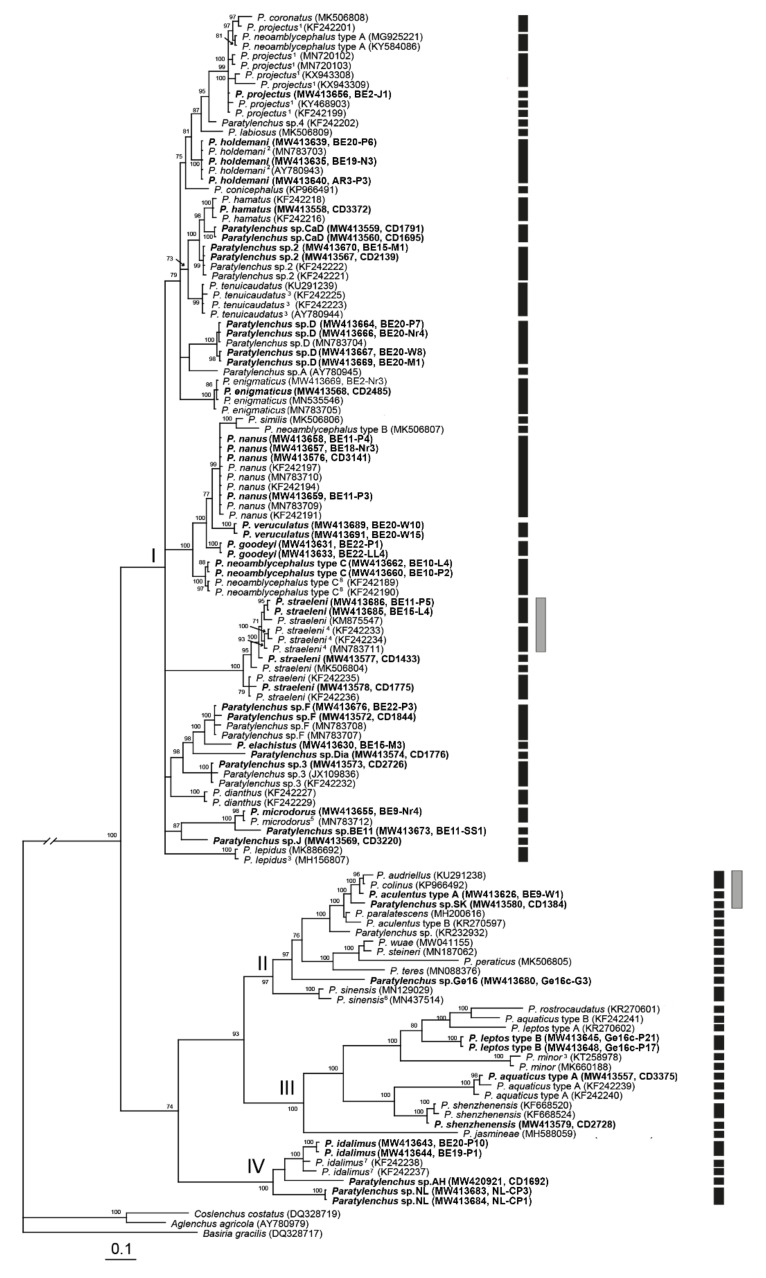
Phylogenetic relationships within populations and species of *Paratylenchus*, as inferred from Bayesian analysis using the D2-D3 of 28S rRNA gene sequence dataset with the GTR + I + G model. Posterior probability of more than 70% is given for the appropriate clades. Newly obtained sequences are indicated in bold. ^1^ = originally identified as *P. nanus*, ^2^ = originally identified as *P. bukowinensis*, ^3^ = originally identified as *Paratylenchus* sp., ^4^ = originally identified as *Paratylenchus* sp.8, ^5^ = originally identified as *Paratylenchus* sp.E, ^6^ = originally identified as *Gracilacus* sp. ^7^ = originally identified as *Paratylenchus* sp.5 and ^8^ = originally identified as *Paratylenchus* sp.6. Black and grey bars represent species boundaries estimated by generalized mixed-yule coalescent (GMYC) and Poisson tree process (bPTP) methods, respectively (only differences with GMYC provided).

**Figure 17 plants-10-00408-f017:**
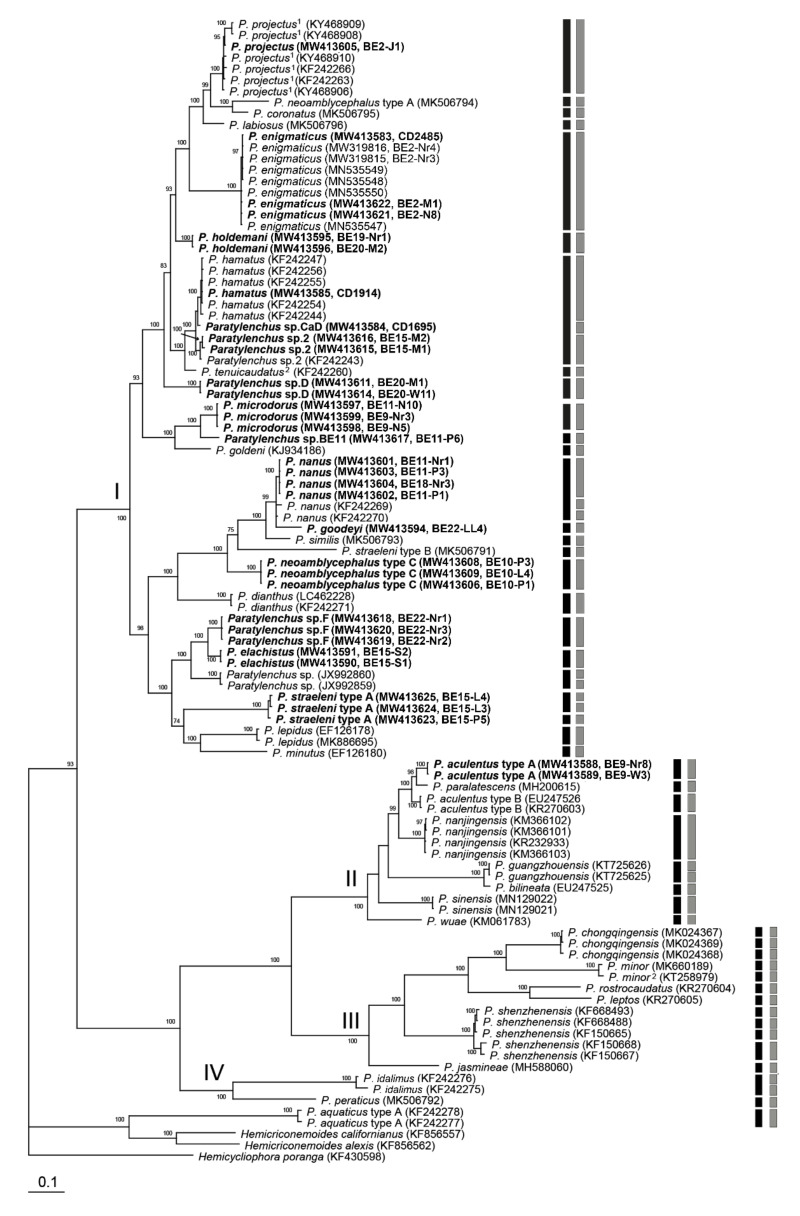
Phylogenetic relationships within populations and species of *Paratylenchus* as inferred from Bayesian analysis using the ITS rRNA gene sequence dataset with the GTR + I + G model. Posterior probability more than 70% is given for appropriate clades. Newly obtained sequences are indicated in bold. ^1^ = originally identified as *P. nanus* and ^2^ = originally identified as *Paratylenchus* sp. Black and grey bars represent species boundaries estimated by GMYC and bPTP methods, respectively.

**Figure 18 plants-10-00408-f018:**
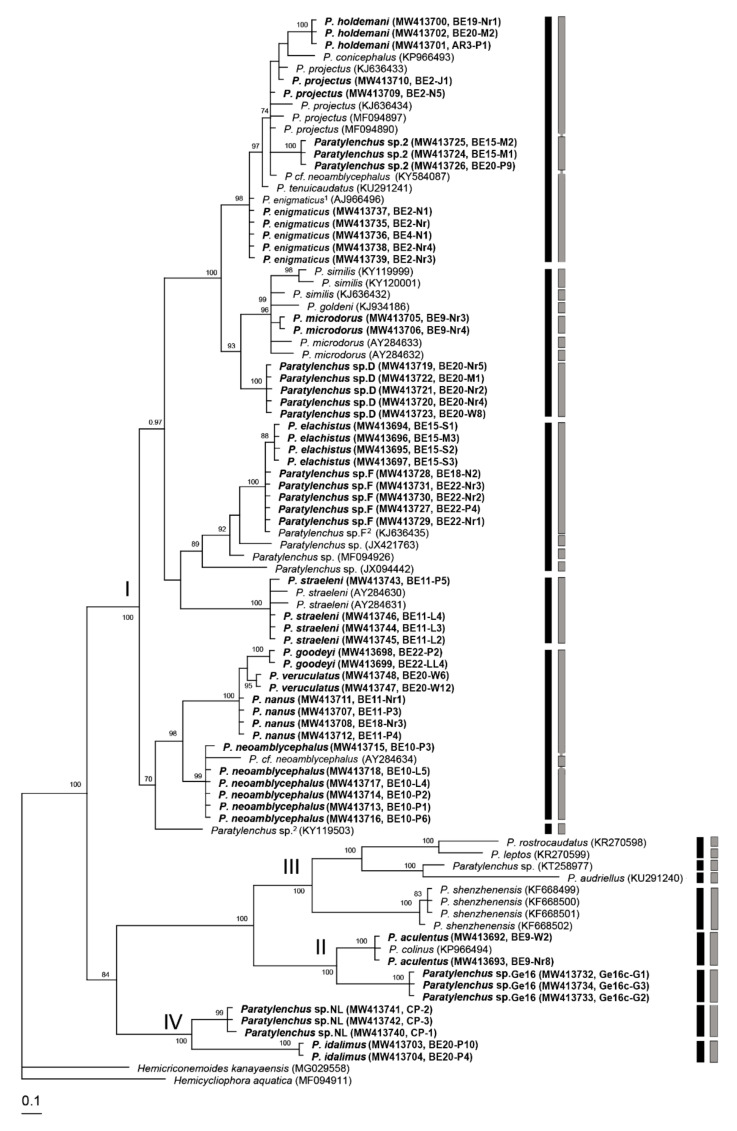
Phylogenetic relationships within populations and species of *Paratylenchus*, as inferred from Bayesian analysis using the 18S rRNA gene sequence dataset with the GTR + I + G model. Posterior probability more than 70% is given for appropriate clades. Newly obtained sequences are indicated in bold. ^1^ = originally identified as *P. dianthus* and ^2^ = originally identified as *P. nanus*. Black and grey bars represent species boundaries estimated by GMYC and bPTP methods, respectively.

**Figure 19 plants-10-00408-f019:**
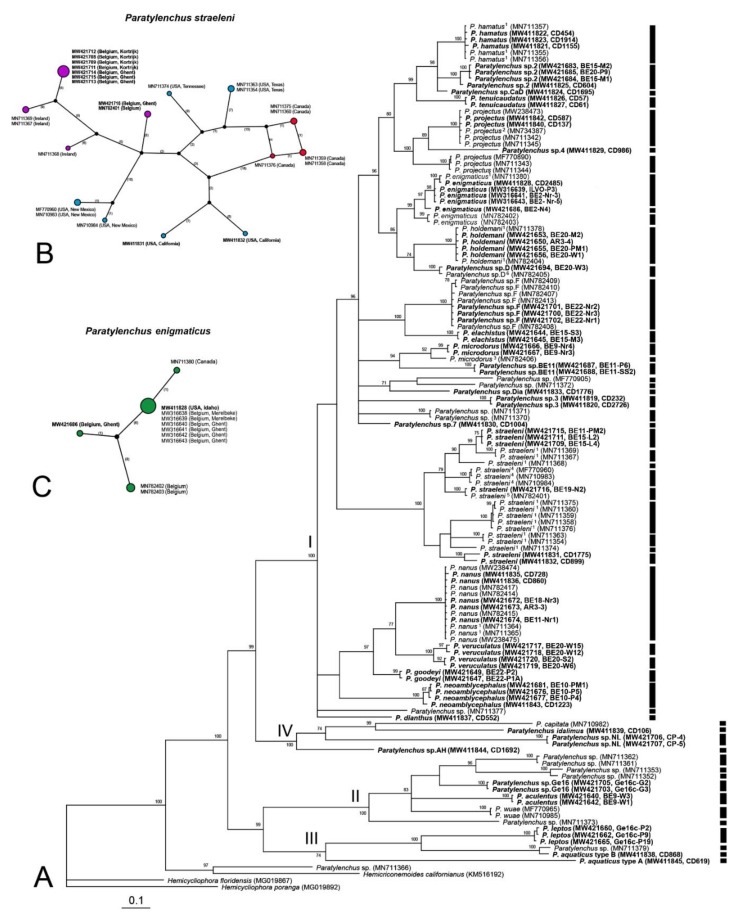
(**A**). Phylogenetic relationships within populations and species of *Paratylenchus*, as inferred from Bayesian analysis using the *COI* gene sequence dataset with the GTR + I + G model. Posterior probability more than 70% is given for appropriate clades. Newly obtained sequences are indicated in bold. ^1^ = originally identified as *Paratylenchus* sp., ^2^ = originally identified as *P. nanus*, ^3^ = identified as *Paratylenchus* sp.E, ^4^ = originally identified as *Gracilacus* sp., ^5^ = originally identified as *Paratylenchus* sp.8, ^6^ = originally identified as *Paratylenchus* sp.B; (**B**). Statistical parsimony network showing the phylogenetic relationships between *COI* haplotypes for *P. straeleni*; (**C**). Statistical parsimony network showing the phylogenetic relationships between *COI* haplotypes for *P*. *enigmaticus*. Pies (circles) represent the sequences with the same haplotype and their size is proportional to the number of these sequences in the samples. Numbers of nucleotide differences between the sequences are indicated on lines connecting the pies. Small black circles represent missing haplotypes. Bars represent species boundaries estimated by both GMYC and bPTP methods (identical results).

**Table 1 plants-10-00408-t001:** List of *Paratylenchus* populations included in this study. Accession numbers of three ribosomal RNA genes (D2-D3 of 28S, ITS and 18S) and a mitochondrial gene (*COI*) fragments are provided for 18 identified and 14 unidentified *Paratylenchus* species. Accession numbers in italics are ones generated in this study.

Species	Locality	Associated Plant Host	Sample Code	GenBank Accession Numbers				Source
				28S rRNA	ITS rRNA	18S rRNA	*COI* of mtDNA	
*P. aculentus*	Belgium, Ghent, Citadel Park; 51°02′05″ N; 3°43′10″ E	Grasses under a tree	BE9	*MW413626–MW41328*	*MW413588–MW413589*	*MW413692–MW413693*	*MW421639–MW421642*	C.M. Etongwe
*P. aquaticus* type A	USA, Florida, Princeton	*Aechmea* sp.	CD3375	*MW413557*	-	-	-	S.A. Subbotin
*P. aquaticus* type A	USA, Hawaii, Waimanalo	Bromeliad (*Neoregelia* sp.)	CD619	KF242239, KF242240	KF242277, KF242278	-	*MW411845*	S.A. Subbotin, Van den Berg et al. [22]
*P. aquaticus* type B	USA, Kansas, Manhattan, WashintonMarlatt park	Grasses	CD868	KF242241	-	-	*MW411838*	S.A. Subbotin, Van den Berg et al. [22]
*P. dianthus*	South Africa, Gauteng, Tarlton	Chrysanthemum	CD552	KF242226–KF242229	KF242271, KF242272	-	*MW411837*	Van den Berg, Van den Berg et al. [22]
*P. enigmaticus*	Belgium, Ghent, Ghent University Botanical Garden; 51°2′7.53″ N; 3°43′20.07″ E	Leek	BE2	-	*MW413621–MW413622*	*MW413735, MW413737–MW413739*	*MW421686*	C.M. Etongwe
*P. enigmaticus*	Belgium, Ghent, Ghent University Botanical Garden; 51°2′7.10″ N; 3°43′19.28″ E	Wild oregano	BE4	-	-	*MW413732–MW413734*	-	C.M. Etongwe
*P. enigmaticus*	USA, Idaho	Unknown plant	CD2485	*MW413568*	*MW413583*	-	*MW411828*	S.A. Subbotin
*P. elachistus*	Belgium, Kortrijk; 50°47′58″ N; 3°11′37″ E	Grasses under a thorny tree	BE15	*MW413629–MW413630*	*MW413590–MW413593*	*MW413694–MW413697*	*MW421643–MW421646*	C.M. Etongwe
*P. goodeyi*	Belgium, Merendree; 51°04′12″ N; 3°34′37″ E	Grasses around a beech tree	BE22	*MW413631–MW413633*	*MW413594*	*MW413698–MW413699*	*MW421647–MW421649*	C.M. Etongwe
*P. hamatus*	USA, California, Merced County, Planada	Fig tree (*Ficus carica*)	CD1155	KF242212	-	-	*MW411821*	S.A. Subbotin, Van den Berg et al. [22]
*P. hamatus*	USA, California	Trees	CD1914	*MW413564*	*MW413585*	-	*MW411823*	S.A. Subbotin
*P. hamatus*	USA, California, Kern county	Grape (*Vitis* sp.)	CD2534a, b	*MW413565, MW413566*	-	-	-	A. Westphal
*P. hamatus*	USA, California, Kern county, Delano	Grape, Cherry	CD3372	*MW413558*	-	-	-	S.A. Subbotin
*P. hamatus*	USA, California, Kern county, Maricopa	Apricot (*Prunus* sp.)	CD454	KF242206, KF242216, KF242217	KF242247, KF242256	-	*MW411822*	S.A. Subbotin, Van den Berg et al. [22]
*P. holdemani*	Belgium, Gouvy, Rogery; 50°14′39.8″ N; 5°57′21.9″ E	Grasses under tree *Fraxinus* sp.	AR3	*MW413636–MW413638, MW413640, MW413642*	-	*MW413701*	*MW421650–MW421652*	P.R. Singh
*P. holdemani*	Belgium, Ghent, Blaarmeersen; 51°02′18.9″ N; 3°41′17.2″ E	Grasses under a thorny tree next to a stream	BE19	*MW413634–MW413635*	*MW413595*	*MW413700*	*MW421658*	C.M. Etongwe
*P. holdemani*	Belgium, Ghent, Blaarmeersen; 51°02′14″ N; 3°41′23″ E	Grasses under a tree	BE20	*MW413639, MW413641*	*MW413596*	*MW413702*	*MW421653–MW421657*	C.M. Etongwe
*P. idalimus*	Belgium, Ghent, Blaarmeersen; 51°02′18.9″ N; 3°41′17.2″ E	Grasses under a thorny tree next to a stream	BE19	*MW413644*	-	-	-	C.M. Etongwe
*P. idalimus*	Belgium, Ghent, Blaarmeersen; 51°02′14″ N; 3°41′23″ E	Grassesundera tree	BE20	*MW413643*	-	*MW413703–MW413704*	-	C.M. Etongwe
*P. idalimus*	USA, California, Napa county, Napa	Grape (*Vitis* sp.)	CD106	KF242237, KF242238	KF242275, KF242276	-	*MW411839*	S.A. Subbotin, Van den Berg et al. [22]
*P. leptos*	Ethiopia, Jimma Zone, Gera district	Coffee	Ge16c	*MW413645–MW413653*	-	-	*MW421659–MW421665*	A.W. Aseffa
*P. microdorus*	Belgium, Zwijnaarde; 51°00′19″ N; 3°42′11″ E	Grasses	BE11	-	*MW413597*	-	-	C.M. Etongwe
*P. microdorus*	Belgium, Ghent, Citadel Park; 51°02′05″ N; 3°43′10″ E	Grasses under a tree	BE9	*MW413654–MW413655*	*MW413598–MW413600*	*MW413705–MW413706*	*MW421666–MW421667*	C.M. Etongwe
*P. nanus*	USA, California, Humboldt county, Trinidad, sample 5B; 41°02′40.6″ N; 124°07′18.1″ W	Unknown plant	CD3141	*MW413576*	-	-	-	S.A. Subbotin
*P. nanus*	USA, Washington, Mason County, Skokomish, sample 32B; 47°18′07.0″ N; 123°10′95.6″ W	Unknown plant	CD3217	*MW413575*	-	-	-	S.A. Subbotin
*P. nanus*	USA, California, Riverside	Grasses	CD728	KF242194, KF242197	KF242267, KF242268	-	*MW411835*	S.A. Subbotin, Van den Berg et al. [22]
*P. nanus*	USA, California, Marin county	*Festuca* sp.	CD850	KF242192, KF242193	-	-	*MW411834*	S.A. Subbotin, Van den Berg et al. [22]
*P. nanus*	USA, California, Marin county	Grasses	CD860	KF242191, KF242195	-	-	*MW411836*	S.A. Subbotin, Van den Berg et al. [22]
*P. nanus*	Belgium, Gouvy, Rogery;50°14′39.8″ N; 5°57′21.9″ E	Grasses under tree *Fraxinus* sp.	AR3	-	-	-	*MW421673*	P.R. Singh
*P. nanus*	Belgium, Zwijnaarde; 51°00′19″ N; 3°42′11″ E	Grasses under a tree	BE11	*MW413658–MW413659*	*MW413601–MW413603*	*MW413707, MW413711–MW413712*	*MW421668–MW421671, MW421674*	C.M. Etongwe
*P. nanus*	Belgium, Ghent, Blaarmeersen; 51°07′14″ N; 2°39′29″ E	Grasses under a tree	BE18	*MW413657*	*MW413604*	*MW413708*	*MW421672*	C.M. Etongwe
*P. neoamblycephalus*	Belgium, Ghent, Citadel Park; 51°02′09″ N; 3°43′06″ E	Cypress tree	BE10	*MW413660–MW413663*	*MW413606–MW413610*	*MW413713–MW413718*	*MW421675–MW421682*	C.M. Etongwe
*P. neoamblycephalus*	USA, California, Madera county, Madera	Grasses	CD1223	KF242190	-	-	*MW411843*	S.A. Subbotin, Van den Berg et al. [22]
*P. projectus*	Belgium, Ghent, Ghent University Botanical Garden; 51°2′7.53″ N; 3°43′20.07″ E	Leek	BE2	*MW413656*	*MW413605*	*MW413709–MW413710*	-	C.M. Etongwe
*P. projectus*	USA, California, Butte county, Gridley	Walnut (*Juglans* sp.)	CD137	KF242199	KF242265, KF242266	-	*MW411840*	S.A. Subbotin, Van den Berg et al. [22]
*P. projectus*	South Africa, Western Cape, George	Bent grass	CD587	KF242198, KF242200	KF242263, KF242264	-	*MW411842*	Van den Berg, Van den Berg et al. [22]
*P. shenzhenensis*	USA, Florida, Apopka	Unknown plant	CD2728	*MW413579*	-	-	-	S.A. Subbotin
*P. straeleni*	Belgium, Zwijnaarde; 51°00′19″ N; 3°42′11″ E	Grasses under a tree	BE11	*MW413686*	*MW413623*	*-*	*MW421713–MW421715*	C.M. Etongwe
*P. straeleni*	Belgium, Kortrijk; 50°47′58″ N; 3°11′37″ E	Grasses under a thorny tree	BE15	*MW413685*	*MW413624–MW413625*	*MW413743–MW413746*	*MW421708–MW421712*	C.M. Etongwe
*P. straeleni*	Belgium, Ghent, Blaarmeersen; 51°02′18.9″ N; 3°41′17.2″ E	Grasses under a thorny tree next to a stream	BE19	-	-	-	*MW421716*	C.M. Etongwe
*P. straeleni*	USA, North Carolina	Unknown plant	CD1433	*MW413577*	-	-	-	W. Ye
*P. straeleni*	USA, California, Monterey county	Oak	CD1775	*MW413578*	-	-	*MW411831*	S.A. Subbotin
*P. straeleni*	USA, California, Napa county	Tree	CD899	KF242236	-	-	*MW411832*	S.A. Subbotin, Van den Berg et al. [22]
*P. tenuicaudatus*	USA, California, Glenn county, Orland	Prune (*Prunus* sp.)	CD57	KF242223, KF242225	KF242261, KF242262	-	*MW411826*	S.A. Subbotin, Van den Berg et al. [22]
*P. tenuicaudatus*	USA, California, Glenn county, Butte City	Prune (*Prunus* sp.)	CD61	KF242224	KF242259, KF242260	-	*MW411827*	S.A. Subbotin, Van den Berg et al. [22]
*P. veruculatus*	Belgium, Ghent, Blaarmeersen; 51°02′14″ N; 3°41′23″ E	Grasses under a tree	BE20	*MW413687–MW413691*	-	*MW413747–MW413748*	*MW421717–MW421722*	C.M. Etongwe
*Paratylenchus* sp.2	Belgium, Kortrijk; 50°47′58″ N; 3°11′37″ E	Grasses under a thorny tree	BE15	*MW413670–MW413671*	*MW413615–MW413616*	*MW413724–MW413725*	*MW421683–MW421684*	C.M. Etongwe
*Paratylenchus* sp.2	Belgium, Ghent, Blaarmeersen; 51°02′14″ N; 3°41′23″ E	Grasses under a tree	BE20	-	-	*MW413726*	*MW421685*	C.M. Etongwe
*Paratylenchus* sp.2	Kyrgyzstan	Trees and grasses	CD2139	*MW413567*	-	-	-	S.A. Subbotin
*Paratylenchus* sp.2	USA, California, Yolo county, Davis	Grasses under a willow tree	CD604	KF242220–KF242222	KF242243	-	*MW411825*	S.A. Subbotin, Van den Berg et al. [22]
*Paratylenchus* sp.3	USA, California, Santa Barbara county, Goleta	Lemon (*Citrus* sp.)	CD232	KF242231, KF242232	KF242273, KF242274	-	*MW411819*	S.A. Subbotin, Van den Berg et al. [22]
*Paratylenchus* sp.3	USA, Florida	Unknown plant	CD2726	*MW413573*	-	-	*MW411820*	S.A. Subbotin
*Paratylenchus* sp.4	USA, Oregon	Trees	CD986	KF242203	-	-	*MW411829*	S.A. Subbotin, Van den Berg et al. [22]
*Paratylenchus* sp.7	USA, California, Riverside, UCR campus	Unknown plant	CD1004	KF242242	-	-	*MW411830*	S.A. Subbotin
*Paratylenchus* sp.AH	USA, California, El Dorado county, Placerville, Apple hills, sample N12	Unknown plant	CD1692	*MW420921*	-	-	*MW411844*	S.A. Subbotin
*Paratylenchus* sp.BE11	Belgium, Zwijnaarde; 51°00′19″ N; 3°42′11″ E	Grasses under a tree	BE11	*MW413672–MW413674*	*MW413617*	-	*MW421687–MW421688*	C.M. Etongwe
*Paratylenchus* sp.CaD	USA, California, El Dorado county, Placerville, Apple hills, sample, N7	Unknown plant	CD1686	*MW413561*	-	-	*MW411841*	S.A. Subbotin
*Paratylenchus* sp.CaD	USA, California, El Dorado county, Placerville, Apple hills, sample N16	Unknown plant	CD1695a, b	*MW413560, MW413562*	*MW413584*	-	*MW411824*	S.A. Subbotin
*Paratylenchus* sp.CaD	USA, California, El Dorado county, Placerville, Apple hills, sample N20	Unknown plant	CD1696	*MW413563*	-	-	-	S.A. Subbotin
*Paratylenchus* sp.CaD	USA, California, Yolo county, Putah Creek	*Rubus* sp.	CD1791	*MW413559*	-	-	-	S.A. Subbotin
*Paratylenchus* sp.D	Belgium, Ghent, Blaarmeersen; 51°02′14″ N; 3°41′23″ E	Grasses under a tree	BE20	*MW413664–MW413669*	*MW413611–MW413614*	*MW413719–MW413723*	*MW421689–MW421699*	C.M. Etongwe
*Paratylenchus* sp.Dia	USA, California, Contra Costa County, Mount Diablo State Park	Unknown plant	CD1776	*MW413574*	-	-	*MW411833*	S.A. Subbotin
*Paratylenchus* sp.F	Belgium, Ghent, Blaarmeersen; 51°07′14″ N; 2°39′29″ E	Grasses under a tree	BE18	-	-	*MW413728*	-	C.M. Etongwe
*Paratylenchus* sp.F	Belgium, Merendree; 51°04′12″ N; 3°34′37″ E	Grasses around a beech tree	BE22	*MW413675–MW413679*	*MW413618–MW413620*	*MW413727, MW413729–MW413731*	*MW421700–MW421702*	C.M. Etongwe
*Paratylenchus* sp.F	Russia, Primorsky Krai, Olginsky district	Unknown plant	CD1842, CD1844	*MW413571, MW413572*	-		-	J. Zograf
*Paratylenchus* sp.Ge16	Ethiopia, Jimma Zone, Gera district	Coffee	Ge16c	*MW413680–MW413682*	-	*MW413732–MW413734*	*MW421703–MW421705*	C.M. Etongwe
*Paratylenchus* sp.J	USA, Washington, Mason County, Skokomish, sample 32E; 47°18′07.0″ N 123°10′95.6″ W	Unknown plant	CD3216	*MW413570*	-	-	-	S.A. Subbotin
*Paratylenchus* sp.J	USA, Oregon, Douglas County, Oakland, sample 35; 43°28′59.9″ N 123°19′24.5″ W	Unknown plant	CD3220	*MW413569*	-	-	-	S.A. Subbotin
*Paratylenchus* sp.NL	The Netherlands, Hilversum	Holly	NL	*MW413683–MW413684*	-	*MW413740–MW413742*	*MW421706–MW421707*	G. Karssen
*Paratylenchus* sp.SK	South Korea	*Pinus* sp.	CD1384	*MW413580*	-		-	S.A. Subbotin

**Table 2 plants-10-00408-t002:** Female morphometrics of *Paratylenchus aculentus*, *Paratylenchus goodeyi*, *Paratylenchus idalimus* and *Paratylenchus straeleni* from fixed specimens mounted in glycerine. All measurements except for ratios and percentages are given in µm and in the form mean ± stdev (range).

Population	*P. aculentus* (BE9)	*P. goodeyi* (BE22)	*P. idalimus* (BE19 and BE20)	*P. straeleni* (BE15)
n	12	17	7	11
L	266 ± 20.1 (233–03)	348 ± 42.5 (266–452)	299 ± 20.7 (278–332)	358 ± 13.1 (330–379)
a	19.6 ± 2.0 (16.3–23.2)	20.7 ± 1.7 (16.7–23.2)	21.0 ± 1.3 (20–23)	22.6 ± 0.9 (20.7–24.3)
b	2.6 ± 0.1 (2.4–2.8)	3.2 ± 0.3 (2.9–3.7)	2 ± 0.2 (2.0–2.4)	3.6 ± 0.1 (3.4–3.7)
c	12.4 ± 1.5 (10.8–15.2)	11.9 ± 1.4 (10.1–13.5)	12.1 ± 0.6 (12.1–13.1)	10.6 ± 1.0 (8.9–11.8)
c	2.8 ± 0.3 (2.4–3.1)	3.0 ± 0.2 (2.8–3.3)	4.0 ± 0.6 (3.1–4.1)	3.4 ± 0.3 (3.0–3.9)
Maximum body width	13.6 ± 1.3 (11.6–15.5)	17.0 ± 3.3 (13.0–27.0)	14.0 ± 1.4 (13.0–16.1)	15.9 ± 0.7 (14.6–16.7)
Stylet length	56.0 ± 3.3 (52.4–61.2)	52.1 ± 2.8 (47.0–58.6)	89.0 ± 3.5 (84.1–93.0)	55.7 ± 1.7 (53.5–58.6)
Cone length	49.1 ± 3.6 (43.0–54.9)	43.0 ± 2.7 (48.2–48.5)	78.0 ± 2.9 (74.0–83.1)	44.7 ± 1.7 (42.2–47.4)
Cone%stylet	87.5 ± 3.8 (80.1–91.0)	82.4 ± 2.5 (78.0–89.6)	88.0 ± 2.2 (83–89)	80 ± 1.8 (76–83)
Knob width	3.2 ± 0.5 (2.3–4.0)	4.1 ± 0.6 (3.3–5.2)	4.0 ± 0.2 (4.0–4.3)	3.9 ± 0.4 (3.1–4.6)
Pharynx length	101 ± 8.3 (87.0–113)	109 ± 11.7 (92.7–133)	130 ± 12.8 (114–147)	100 ± 3.8 (92.1–105)
Anterior end to SE pore	66.7 ± 5.2 (54.3–74.4)	80.7 ± 8.6 (68.5–99.0)	93.0 ± 11.7 (82.0–115)	82.5 ± 2.9 (79.4–87.6)
SE pore%L	25.2 ± 0.9 (23.3–26.4)	22.9 ± 1.4 (21.0–25.8)	31 ± 4.2 (28–40)	23 ± 0.7 (22–24)
Anterior end to vulva	193 ± 16.1 (165–218)	279 ± 31.9 (216–356)	233 ± 17.9 (214–260)	270 ± 18.8 (249–330)
V%	72.5 ± 1.5 (70.8–75.7)	80.1 ± 1.4 (77.8–82.3)	78.0 ± 0.9 (77–79)	81 ± 1.9 (80–84)
Body width at anus	7.6 ± 0.5 (7.0–8.3)	9.7 ± 0.5 (9.7–10.0)	7.0 ± 0.1 (7.0–7.1)	10.1 ± 0.4 (9.1–10.5)
Tail length	20.9 ± 2.3 (18.1–25.1)	29.0 ± 2.4 (25.8–32.1)	25.0 ± 2.2 (22.0–28.1)	34.4 ± 3.7 (31–40.8)

**Table 3 plants-10-00408-t003:** Female morphometrics of *Paratylenchus elachistus*, *Paratylenchus holdemani*, *Paratylenchus microdorus* and *Paratylenchus veruculatus* from fixed specimens mounted in glycerine. All measurements except for ratios and percentages are given in µm and in the form mean ± stdev (range).

Population	*P. elachistus* (BE15)	*P. holdemani* (AR3)	*P. microdorus* (BE9)	*P. veruculatus* (BE20)
n	24	31	10	15
L	301 ± 12.5 (283–329)	359 ± 47 (285–475)	330.7 ± 20 (297–355)	286 ± 24.7 (251–331)
a	20.4 ± 1.1 (17.7–22.6)	20.9 ± 1.9 (16.4–25.2)	21.4 ± 1.4 (18.7–23.1)	19.8 ± 1.9 (17.2–23.3)
b	4.1 ± 0.2 (3.8–4.3)	4.1 ± 0.7 (2.2–5.1)	5.0 ± 0.5 (4.5–6.2)	3.8 ± 0.3 (3.3–4.2)
c	12.1 ± 0.9 (10.9–14.3)	14.8 ± 1.4 (12.4–17.7)	10.6 ± 0.9 (9.2–12)	17.8 ± 1.8 (14.6–20.6)
c’	2.8 ± 0.2 (2.4–3.3)	2.5 ± 0.3 (2.1–3.2)	3.8 ± 0.5 (2.8–4.5)	2.2 ± 0.2 (1.8–2.6)
Maximum body width	14.8 ± 1.0 (12.7–16.6)	17.3 ± 3.0 (11.3–23.8)	15.5 ± 1.4 (13.3–17.1)	14.5 ± 1.4 (12.5–16.5)
Stylet length	20.9 ± 0.7 (19.7–22.2)	22.5 ± 2.0 (19.0–26.1)	12.4 ± 1.3 (10.6–14.7)	14.2 ± 0.5 (13.1–14.8)
Cone length	13.3 ± 0.4 (12.4–13.9)	15.1 ± 1.1 (13.2–18.5)	6.7 ± 1.3 (4.8–8.1)	8.9 ± 0.3 (8.3–9.3)
Cone%stylet	63.7 ± 1.5 (60.5–67.5)	67.3 ± 3.5 (60.9–77.4)	53.4 ± 5.7 (45.3–60.4)	62.8 ± 1.1 (60.3–64.8)
Knob width	3.5 ± 0.2 (3.1–4.1)	3.3 ± 0.4 (2.9–4.2)	-	3.1 ± 0.3 (2.7–3.5)
Pharynx length	74.5 ± 2.6 (70.4–80.6)	89.7 ± 21.5 (66.1–161)	66.6 ± 6.4 (56.2–76.1)	75.8 ± 7.0 (60.8–88.4)
Anterior end to SE pore	60.7 ± 3.9 (54.0–68.5)	74.8 ± 9.1 (60.1–99.0)	63.6 ± 4.7 (57.5–71.3)	62.3 ± 6.5 (51.2–74.4)
SE pore%L	20.1 ± 0.9 (18.6–21.9)	21.2 ± 1.8 (16.4–23.7)	19.2 ± 1.1 (17.3–20.6)	21.8 ± 2.0 (17.5–25.6)
Anterior end to vulva	245 ± 10.2 (226–269)	303 ± 40.9 (238–391)	-	245 ± 22.0 (215–284)
V%	81.3 ± 0.9 (79.7–83.2)	84.3 ± 1.8 (81.3–90.5)	81 ± 1.6 (79.1–82.8)	85.7 ± 1.4 (83.8–89.7)
Body width at anus	8.9 ± 0.7 (7.5–9.9)	10.0 ± 1.3 (7.2–12.3)	8.5 ± 1.3 (6.8–10.8)	7.3 ± 0.6 (6.3–8.8)
Tail length	24.8 ± 2.2 (20.9–29.1)	25.2 ± 2.8 (20.0–29.5)	31.8 ± 3.1 (28.1–35.9)	16.1 ± 1.7 (13.6–19.1)

**Table 4 plants-10-00408-t004:** Female morphometrics of *Paratylenchus nanus*, *Paratylenchus neoamblycephalus*, *Paratylenchus* sp.2, *Paratylenchus* sp.D and *Paratylenchus* sp.F from fixed specimens mounted in glycerine. All measurements except for ratios and percentages are given in µm and in the form mean ± stdev (range).

Population	*P. nanus* (BE11)	*P. neoamblycephalus* (BE10)	*Paratylenchus* sp.2 (BE15)	*Paratylenchus* sp.D (BE20)	*Paratylenchus* sp.F (BE22)
n	30	15	16	11	17
L	318 ± 15.8 (287–352)	337 ± 20.2 (301–367)	347 ± 20.7 (308–389)	328 ± 36.1 (285–387)	300 ± 21.1 (264–339)
a	18.1 ± 1.3 (15.6–20.4)	18.4 ± 1.0 (16.8–19.9)	23.2 ± 1.8 (20.1–28.7)	21.5 ± 1.6 (19.1–24.4)	20.3 ± 1.3 (18.4–23.0)
b	3.9 ± 0.3 (3.5–4.8)	4.3 ± 0.5 (3.4–4.9)	3.9 ± 0.3 (3.5–4.4)	3.6 ± 0.3 (3.3–4.1)	4.0 ± 0.3 (3.7–4.6)
c	14.8 ± 1.3 (12.8–16.8)	14.6 ± 1.5 (12.9–16.6)	13.2 ± 0.7 (12.3–14.4)	14.5 ± 1.4 (12.0–15.8)	12.9 ± 0.8 (11.8–14.0)
c’	2.0 ± 0.2 (1.7–2.6)	2.2 ± 0.2 (1.9–2.4)	3.1 ± 0.2 (2.9–3.5)	2.7 ± 0.3 (2.5–3.2)	2.6 ± 0.2 (2.3–2.9)
Max. body width	17.6 ± 1.2 (15.4–20.5)	18.3 ± 1.4 (16.8–20.7)	15.1 ± 1.3 (13.2–16.8)	15.3 ± 2.2 (13.2–19.9)	14.8 ± 1.1 (13.3–16.6)
Stylet length	28.8 ± 1.2 (26.7–31.2)	33.4 ± 0.9 (32.0–34.3)	28.4 ± 1.5 (26.5–31.4)	27.5 ± 1.0 (25.7–28.9)	27.6 ± 1.2 (25.3–29.6)
Cone length	20.2 ± 1.4 (17.7–23.4)	22.6 ± 0.9 (21.4–24.5)	19.2 ± 1.0 (17.5–20.8)	17.7 ± 0.8 (17.0–19.2)	18.5 ± 0.9 (17.1–20.3)
Cone%stylet	70.1 ± 3.6 (64.9–78.5)	67.8 ± 2.8 (63.0–72.5)	67.5 ± 2.0 (64.4–71.2)	64.3 ± 2.0 (60.7–67.3)	67.1 ± 1.3 (65.2–69.5)
Knob width	3.9 ± 0.4 (3.1–4.6)	4.8 ± 0.2 (4.4–5.1)	4.1 ± 0.3 (3.5–4.7)	4.0 ± 0.3 (3.5–4.6)	3.6 ± 0.3 (3.3–4.1)
Pharynx length	81.6 ± 5.4 (65.4–91.1)	79.7 ± 9.1 (65.7–93.8)	88.2 ± 4.1 (78.0–96.7)	89.6 ± 7.3 (76.0–104)	74.8 ± 4.9 (67.7–83.1)
Ant. end to SE pore	65.7 ± 6.0 (54.7–75.0)	63.9 ± 4.7 (52.2–70.0)	72.3 ± 4.5 (64.2–90.0)	74.6 ± 6.8 (66.7–90.2)	63.0 ± 5.4 (51.5–70.6)
SE pore%L	20.6 ± 1.4 (17.0–22.5)	19.0 ± 1.5 (15.8–21.7)	20.8 ± 1.3 (19.4–23.5)	23.6 ± 0.9 (22.1–24.8)	21.0 ± 1.6 (18.6–24.2)
Ant. end to vulva	270 ± 18.8 (249–330)	276 ± 15.0 (247–296)	286 ± 17.1 (252–313)	270 ± 27.9 (239–320)	217 ± 17.2 (217–278)
V%	83.8 ± 1.1 (81.7–85.7)	81.9 ± 0.9 (80.7–83.8)	82.2 ± 0.8 (81.2–83.5)	83.7 ± 1.0 (81.8–85.2)	82 ± 0.7 (80.9–83.5)
Body width at anus	10.9 ± 0.9 (9.3–12.6)	10.6 ± 0.8 (9.4–12.0)	8.6 ± 0.7 (7.6–9.8)	8.0 ± 0.6 (7.3–8.7)	9.0 ± 0.6 (8.0–9.9)
Tail length	21.7 ± 1.9 (18.9–25.5)	23.2 ± 2.6 (20.2–27.8)	26.1 ± 2.0 (23.0–28.7)	22.0 ± 3.0 (18.0–25.6)	23.1 ± 2.4 (20.0–26.5)

**Table 5 plants-10-00408-t005:** List of some existing unidentified or incorrectly classified *Paratylenchus* sequences on the GenBank reassigned to corrected species. In total, 18 D2-D3 of 28S, 3 ITS, 3 18S rRNA and 25 *COI* gene sequences have been reassigned.

Gene	GenBank Accession No.	Linked Species	Country of Origin	Reference	Reassigned Species Name
D2-D3	MN437514	*Gracilacus* sp.	Myanmar	Du, Y. (Unpublished)	*P. sinensis*
D2-D3	AY780943	*P. bukowinensis*	Italy	Subbotin et al. [37]	*P. holdemani*
D2-D3	MN088372	*P. bukowinensis*	Iran	Mirbabaei et al. [46]	*P. holdemani*
D2-D3	MN783703	*P. bukowinensis*	Belgium	Etongwe et al. [47]	*P. holdemani*
D2-D3	AY780944	*Paratylenchus* sp.	Italy	Subbotin et al. [37]	*P. tenuicaudatus*
D2-D3	MH156807	*Paratylenchus* sp.	China	Fan et al. (Unpublished)	*P. lepidus*
D2-D3	KF242223	*Paratylenchus* sp.1	USA	Van den Berg et al. [22]	*P. tenuicaudatus*
D2-D3	KF242224	*Paratylenchus* sp.1	USA	Van den Berg et al. [22]	*P. tenuicaudatus*
D2-D3	KF242225	*Paratylenchus* sp.1	USA	Van den Berg et al. [22]	*P. tenuicaudatus*
D2-D3	KF242237	*Paratylenchus* sp.5	USA	Van den Berg et al. [22]	*P. idalimus*
D2-D3	KF242238	*Paratylenchus* sp.5	USA	Van den Berg et al. [22]	*P. idalimus*
D2-D3	KT258978	*Paratylenchus* sp.	China	Liu et al. (Unpublished)	*P. minor*
D2-D3	KF242189	*Paratylenchus* sp.6	USA	Van den Berg et al. [22]	*P. neoamblycephalus*
D2-D3	KF242190	*Paratylenchus* sp.6	USA	Van den Berg et al. [22]	*P. neoamblycephalus*
D2-D3	KF242233	*Paratylenchus* sp.8	USA	Van den Berg et al. [22]	*P. straeleni*
D2-D3	KF242234	*Paratylenchus* sp.8	USA	Van den Berg et al. [22]	*P. straeleni*
D2-D3	MN783711	*Paratylenchus* sp.8	Belgium	Etongwe et al. [47]	*P. straeleni*
D2-D3	MN783712	*Paratylenchus* sp.E	Belgium	Etongwe et al. [47]	*P. microdorus*
ITS	KT258979	*Paratylenchus* sp.	China	Liu et al. (Unpublished)	*P. minor*
ITS	KF242260	*Paratylenchus* sp.1	USA	Van den Berg et al. [22]	*P. tenuicaudatus*
ITS	KF242259	*Paratylenchus* sp.1	USA	Van den Berg et al. [22]	*P. tenuicaudatus*
18S	AJ966496	*P. dianthus*	Belgium	Meldal et al. [65]	*P*. *enigmaticus*
18S	KJ636435	*P. nanus*	The Netherlands	Van Megen et al. (Unpublished)	*Paratylenchus* sp.F
18S	KY119503	*P. nanus*	Ireland	Ortiz et al. [66]	*Paratylenchus* sp.
*COI*	MF770960	*Gracilacus* sp.	USA	Munawar et al. (Unpublished)	*P. straeleni*
*COI*	MN710983	*Gracilacus* sp.	USA	Powers et al. [49]	*P. straeleni*
*COI*	MN710984	*Gracilacus* sp.	USA	Powers et al. [49]	*P. straeleni*
*COI*	MN711354	*Paratylenchus* sp.	USA	Powers et al. [49]	*P. straeleni*
*COI*	MN711355	*Paratylenchus* sp.	USA	Powers et al. [49]	*P. hamatus*
*COI*	MN711356	*Paratylenchus* sp.	USA	Powers et al. [49]	*P. hamatus*
*COI*	MN711357	*Paratylenchus* sp.	USA	Powers et al. [49]	*P. hamatus*
*COI*	MN711358	*Paratylenchus* sp.	Canada	Powers et al. [49]	*P. straeleni*
*COI*	MN711359	*Paratylenchus* sp.	Canada	Powers et al. [49]	*P. straeleni*
*COI*	MN711360	*Paratylenchus* sp.	Canada	Powers et al. [49]	*P. straeleni*
*COI*	MN711363	*Paratylenchus* sp.	USA	Powers et al. [49]	*P. straeleni*
*COI*	MN711367	*Paratylenchus* sp.	Ireland	Powers et al. [49]	*P. straeleni*
*COI*	MN711368	*Paratylenchus* sp.	Ireland	Powers et al. [49]	*P. straeleni*
*COI*	MN711369	*Paratylenchus* sp.	Ireland	Powers et al. [49]	*P. straeleni*
*COI*	MN711374	*Paratylenchus* sp.	USA	Powers et al. [49]	*P. straeleni*
*COI*	MN711375	*Paratylenchus* sp.	Canada	Powers et al. [49]	*P. straeleni*
*COI*	MN711376	*Paratylenchus* sp.	Canada	Powers et al. [49]	*P. straeleni*
*COI*	MN711378	*Paratylenchus* sp.	Poland	Powers et al. [49]	*P. holdemani*
*COI*	MN711380	*Paratylenchus* sp.	Canada	Powers et al. [49]	*P*. *enigmaticus*
*COI*	MN711364	*Paratylenchus* sp.	Ireland	Powers et al. [49]	*P. nanus*
*COI*	MN711365	*Paratylenchus* sp.	Ireland	Powers et al. [49]	*P. nanus*
*COI*	MN782401	*Paratylenchus* sp.8	Belgium	Etongwe et al. [47]	*P. straeleni*
*COI*	MN782404	*Paratylenchus* sp.B	Belgium	Etongwe et al. [47]	*P. holdemani*
*COI*	MN782405	*Paratylenchus* sp.B	Belgium	Etongwe et al. [47]	*Paratylenchus* sp.D
*COI*	MN782406	*Paratylenchus* sp.E	Belgium	Etongwe et al. [47]	*P. microdorus*

## Data Availability

The datasets generated during and/or analyzed during the currentstudy are available from the corresponding author on reasonable request.

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
