# Peer review of "Integrative Taxonomy and Molecular Phylogeny of the Plant-Parasitic Nematode Genus Paratylenchus (Nematoda: Paratylenchinae): Linking Species with Molecular Barcodes"

_plants, 2021, doi:10.3390/plants10020408_

Round 1
Reviewer 1 Report
This paper combines morphological identification with molecular phylogeny to classify populations of the plant-parasitic nematode genus Paratylenchus.
Clear and thorough morphological descriptions and accompanying light and scanning electron microscopy images are provided for 10 species of Paratylenchus, as well as four unidentified species within this genus.
The paper makes a valuable contribution to nematode taxonomy.
The following minor, mainly grammatical, edits were noted:
- Line 22: delete ‘new’ and insert on line 23 before ‘gene sequences’.
- Line 28: replace the word ‘reassignation’ with ‘reassignment’.
- More than 10 keywords are provided.
- Line 36: Start the sentence with ‘The plant-parasite nematode genus. . .’ (Replace ‘This’ with ‘The’)
- Line 72: change ‘started molecularly characterising. . .’ to ‘started to molecularly characterise. . .’
- Line 72: ‘respectively’ should be placed after the two lists are given (authors and gene sequences). Remove ‘respectively from line 72 and insert after ‘sequences’ on line 73.
- Line 78: replace ‘. . .updated with’ with ‘. . .updated by’.
- Line 82: replace ‘work more integrative and to include’ with ‘integrate’
- Line 101: replace the word ‘reassignation’ with ‘reassignment’.
- Line 105: replace the word ‘Deliminting’ with ‘Delimination’
- Line 106, 107, 113: Change tense in results section from present to past tense. Replace ‘are’ with ‘were’ and ‘is’ with ‘was’
- Line 140, 142: italicise species names
- Table 1: Present Table 1 as a supplementary table rather than in the article. Add an extra column heading title ‘Accession numbers’ merged over the over the four columns of gene sequences.
- Line 155: Italicise ‘En face’
- The order that the morphometrics of the species are presented in Table 2, 3 and 4 should follow the alphabetical order that the morphological descriptions are presented.
- Line 228: Delete ‘before’ and insert ‘previously’ . . . .‘and reported previously from Czech Republic. . .’
- Line 236: If the stylet length is shorter, then it is given that the previously reported P. bukowinensis is above the value given. Delete ‘with above 22.5 µm.’
- Line 284: Insert ‘rRNA’ between ‘18S’ and ‘sequences’
- Line 487: hyphenate ‘secretory-excretory’ pore.
- Line 539: move ‘respectively’ to the end of the sentence.
- Line 592: replace ‘Taken’ with ‘Taking’
- Table 5: replace column heading ‘Corrected species name’ with ‘Reassigned species name’
- Line 600: move the position of ‘valid’ to before species. . . ‘with 124 valid species’
- Line 612: Write 72% in words as it is the start of the sentence.
25. Line 624: move the position of ‘respectively’ to after ‘species’.
- Line 671: remove the word ‘Collective’.
- Line 703: replace ‘of’ with ‘from’
- Line 705: Insert comma after ‘lysis’
- Line 705: insert ‘as’ . . . .’was also used as a nematode. . .
- Line 753: change ‘Strong link’ to ‘Strong links’
Reviewer 2 Report
Novel information related with the taxonomy and phylogeny of the genus Paratylenchus enhanced the quality and scientific relevance of the work.
See comments and suggestions on the manuscript.
Novel information related with the taxonomy and phylogeny of the genus Paratylenchus enhanced the quality and scientific relevance of the work.
See comments and suggestions on the manuscript.
I would suggest that the manuscript can be accepted after minor revision.
Reviewer 3 Report
The work by Singh et al. is well written and has, overall, a great flow of ideas. The study is based on an intensive/integrative approach including morphology and DNA sequences from multiple populations/species collected worldwide. It clarifies/corrects multiple Paratylenchus DNA barcodes previously published in GenBank as well as provides numerous new DNA barcodes for multiple species in this PPN genus. The reviewer has made minor changes/suggestions along the text that the authors should revisit prior to submitting the manuscript. Addition to those comments, the reviewer has the following comments:
General
1- rRNA vs. rDNA: please, adopt only one (see Table 1 headers).
2- English style: be consistent throughout the text (e.g. characterization vs. characterisation).
3- Species description: why some morphological features are included in some and not in others (e.g., body length, deirids, etc.). For consistence, adopt the same style/set of morphological features in each description.
4- Unidentified and incorrectly identified. The authors often use these words in the same sentence which sometimes makes the text a bit confusing. Try to use other synonyms when these are too close in the sentence.
5- Some (or all) of the information on the molecular characterization of the species could be included on Table 1 (e.g. number of Barcodes, Blast match, intraspecfic divergence, etc.), this would remove/reduce some of the redundant text in the manuscript.
Major comments:
1- The authors talk about highly and weakly supported clades on their phylogenetic trees, but do not provide a definition or a cutoff (i.e. BPP value) for it. Also, little is discussed about the branching patterns within each clade across genes.
2- The authors use different phylogenetic approaches in the study, but did not discuss the monophyly, and more importantly the lack of, in multiple Paratylenchus species. For the latter issue, is there evidence for the synonimization of Paratylenchus species?
3- How the species delineation methods impacted the results (i.e., the mechanics of each method)? Also, how the genes used may have affected the results (conserved vs. variable)?
4- It is surprising that with all this molecular data, the authors did not intent to map, or at least discuss, some of the relevant morphological characters that may support phylogenetic relationships within Paratylenchus (e.g., is the presence/absence of deirid important).
